


# Review article: Natural hazard risk assessments at the global scale

Philip J. Ward[1], Veit Blauhut[2], Nadia Bloemendaal[1], James E. Daniell[3,4,5], Marleen C. de Ruiter[1], Melanie Duncan[6], Robert Emberson[7], Susanna F. Jenkins[8], Dalia Kirschbaum[7], Michael Kunz[5,9], Susanna Mohr[5,9], Sanne Muis[1,10], Graeme Riddell[11], Andreas Schäfer[3,9], Thomas Stanley[7,12,13], Ted I.E. Veldkamp[1], Hessel C. Winsemius[10,14]

1. Institute for Environmental Studies (IVM), Vrije Universiteit Amsterdam, De Boelelaan 1087, 1081 HV Amsterdam, The Netherlands
2. Environmental Hydrological Systems, University of Freiburg, Germany
3. Geophysical Institute (GPI), Karlsruhe Institute of Technology (KIT), Karlsruhe, Germany
4. Risklayer GmbH, Karlsruhe, Germany
5. Center for Disaster Management and Risk Reduction Technology, KIT, Karlsruhe, Germany
6. British Geological Survey, The Lyell Centre, Edinburgh, UK
7. NASA Goddard Space Flight Center, Greenbelt, Maryland, USA
8. Earth Observatory of Singapore, Asian School of the Environment, Nanyang Technological University, Singapore
9. Institute of Meteorology and Climate Research (IMK-TRO), KIT, Karlsruhe, Germany
10. Deltares, Boussinesqweg 1, 2629 HV Delft, The Netherlands
11. School of Civil, Environmental and Mining Engineering, The University of Adelaide, Adelaide, Australia
12. Universities Space Research Association, Columbia, Maryland, USA
13. Goddard Earth Sciences Technology and Research, Columbia, Maryland, USA
14. TU Delft, PO Box 5048, 2600 GA Delft, The Netherlands

*Correspondence to*: Philip J. Ward (philip.ward@vu.nl)

**Abstract.** Since 1990, natural hazards have led to over 1.6 million fatalities globally, and economic losses are estimated at an average of around $260-310 billion per year. The scientific and policy community recognise the need to reduce these risks. As a result, the last decade has seen a rapid development of global models for assessing risk from natural hazards at the global scale. In this paper, we review the scientific literature on natural hazard risk assessments at the global scale, and specifically examine whether and how they have examined future projections of hazard, exposure, and/or vulnerability. In doing so, we examine similarities and differences between the approaches taken across the different hazards, and identify potential ways in which different hazard communities can learn from each other. For example, we show that global risk studies focusing on hydrological, climatological, and meteorological hazards, have included future projections and disaster risk reduction measures (in the case of floods), whilst these are missing in global studies related to geological hazards. The methods used for projecting future exposure in the former could be applied to the geological studies. On the other hand, studies of earthquake and tsunami risk are now using stochastic modelling approaches to allow for a fully probabilistic assessment of risk, which could benefit the modelling of risk from other hazards. Finally, we discuss opportunities for learning from methods and approaches being developed and applied to assess natural hazard risks at more continental or regional scales. Through this paper, we hope to encourage dialogue on knowledge sharing between scientists and communities working on different hazards and at different spatial scales.



## 1. Introduction

The risk caused by natural hazards is extremely high and increasing. Since 1990, reported disasters have led to over 1.6 million fatalities globally, and economic losses are estimated at an average of around $260-310 billion per year (UNDRR, 2015a). The need to reduce the risk associated with natural hazards is recognised by the international community, and is at the heart of the Sendai Framework for Disaster Risk Reduction (Sendai Framework; UNDRR, 2015b). The Sendai Framework adopts the conceptualisation of disaster risk as the product of hazard, exposure, and vulnerability. The hazard refers to the hazardous

phenomena itself, such as a flood event, including its characteristics and probability of occurrence; exposure refers to the location of economic assets or people in a hazard-prone area; and vulnerability refers to the susceptibility of those assets or people to suffer damage and loss (e.g. due to unsafe housing and living conditions, or lack of early warning procedures). Reducing risk is also recognised as a key aspect of sustainable development in the Sustainable Development Goals (SDGs) and the Paris Agreement on climate change.

Managing disaster risk requires an understanding of risk and its drivers, from household to global scales. This includes an understanding of how risk may change in the future, and how that risk may be reduced through disaster risk reduction (DRR) efforts. A global scale understanding of disaster risk is important for identifying regions most at risk, providing science-based information for DRR advocacy, and assessing the potential effectiveness of DRR solutions. Efforts to assess and map natural hazard risk at the global scale have been ongoing since the mid 2000s, starting with the Natural Disaster Hotspots analysis of

Dilley et al. (2005). This was followed by the global risk assessments for an increasing number of natural hazards in the biennial Global Assessment Reports (GARs) of the United Nations Office for Disaster Risk Reduction (UNDRR) (UNDRR, 2009, 2011, 2013, 2015a, 2017).

At the same time, there have been increasing efforts in the scientific community to develop global methods to assess the potential risks of natural hazards at the global scale, in order to inform (inter)national decision-makers. In 2012, the session

Global and continental scale risk assessment for natural hazards: methods and practice was established at the General Assembly of the European Geosciences Union, in order to bring together people and institutes working on large scale risk assessment from different disciplinary communities. The session has been held each year since 2012, leading to the current special issue in Natural Hazards and Earth System Sciences. One of the enduring themes of these sessions has been assessing future natural hazard risk. In Figure 1, we show the percentage of abstracts accepted to this session each year that explicitly mentions

examining future projections of risk based on future scenarios of hazard, exposure, or vulnerability projections. The number of abstracts dealing with future scenarios of hazard and exposure is much higher than those dealing with future scenarios of vulnerability. Over the eight year period that the session has run, scenarios of future hazards have been the most common aspect amongst those abstracts dealing with future risk projections.

**Figure 1**



In this paper, we review the scientific literature on natural hazard risk assessments at the global scale, and specifically examine whether and how they have examined future projections of hazard, exposure, and/or vulnerability. In doing so, we examine similarities and differences between the approaches taken across the different hazards, thereby identifying potential ways in which different hazard communities can learn from each other whilst acknowledging the challenges faced by the respective

hazard communities. First, we review the scientific literature for each natural hazard risk individually. Second, we compare and contrast the state of the art across the different hazard types. Third, we conclude by discussing future research challenges faced by the global risk modelling community, and several opportunities for addressing those challenges.

## 2.  Review of literature per hazard type

In this section, we review scientific literature on global scale natural hazard risk assessments. We limit the review to those

studies that have used a spatial representation of the risk elements. Therefore, we do not include studies that use global (or continental) damage functions to directly translate from a global stressor (e.g. global temperature change) to a loss, or studies that solely use normalisation methods to assess trends in past reported losses. The results are presented for major natural hazards, including those that have been modelled for the UNDRR Global Assessment Reports.

In carrying out the review, we focus on the aspects described in the bullets below. For each of the reviewed studies, the

information across these aspects is summarised in Table 1.

- *Risk elements*: we indicate whether hazard, exposure, and vulnerability are explicitly represented in the study. If so, we indicate whether these are represented in a static or dynamic nature over time. By static, we mean that no future projections are included (i.e only current representation is used), and by dynamic we mean that future projections are included. In the table, no representation is shown in white, static in orange, and dynamic in green.

- *Resolution of risk elements*: we indicate the spatial resolution at which each risk element is represented.

- *Risk indicators*: we show the indicators used to express the risk.

- *Future DRR measures*: we indicate whether the study explicitly represents future DRR measures in its modelling framework, and if so we indicate whether these are related to structural, nature-based, or non-structural measures. We also indicate whether the costs of these measures are assessed, and whether the impact of human behaviour on their

effectiveness is assessed.

- *Risk analysis*: we indicate the type of risk assessment that was carried out. We have classed these as either non-probabilistic (NP) or probabilistic (P). By probabilistic, we mean that expected annual impacts are assessed either by integrating across return periods or based on a probabilistic stochastic event set. We also indicate whether the risk assessment represents hazard using stochastic event sets (S), return period maps (R), maps of yearly (Y) or monthly

(M) hazard, past events (PE), or the radius around a specific volcano (V). We also indicate the time horizon reported in the study, and the resolution at which the risk analysis is carried out, and the geographical scale to which the results are aggregated.





In the following subsections, each hazard is reported individually.

**Table 1**

### 2.1. Floods

### 2.1.1. River floods

A relatively large number of studies have been carried out to assess global risk from river floods. A description of each study and main findings can be found in the Supplementary Information. These studies are summarised in this section, and details

of the reviewed aspects for each study are shown in Table 1.

Dilley et al. (2005) was the first study to overlay regions that had been affected by large floods between 1985-2003 with population data. Güneralp et al. (2015) used the same data to map potential changes in urban areas in flood-prone regions. However, the flood maps used only provide a general impression of regions affected by floods, but not necessarily actual inundated areas, and are therefore not included in Table 1. The earliest assessments of future river flood risk examine changes

in the number of people experiencing discharge flows of different magnitudes, at coarse resolutions ranging from 0.5° x 0.5° to 'large river basins' (Kleinen and Petschel-Held, 2007; Hirabayashi and Kanae, 2009; Arnell and Lloyd-Hughes, 2014; Arnell and Gosling, 2016). Whilst the numbers differ greatly between studies, they all reveal large increase throughout the 20th century.

Since then, river flood risk modelling has progressed to examine flood hazard based on modelled inundation maps at

resolutions varying from 30" x 30" to 2.5' x 2.5'. Several early studies only examined current risk (Ward et al., 2013; UNDRR, 2015a) or examined either dynamic hazard (Hirabayashi et al., 2013; Alfieri et al., 2017; Willner et al., 2018) or dynamic exposure (Jongman et al., 2012). Several of the most recent studies have used dynamic projections of both hazard and exposure (Winsemius et al., 2016; Ward et al., 2017; Dottori et al., 2018), whilst Jongman et al. (2015) also added dynamic vulnerability. In terms of hazard modelling, there has been a clear movement from the coarse resolution maps of extreme discharge to

approaches using inundation maps of different return periods, whereby probabilistic risk is also assessed. There has been an overall shift from coarse exposure maps (0.5° x 0.5° to 'large river basins') of population only, to higher resolution maps with population, GDP, and land use. This has accompanied a transition from addressing affected people only, to moving towards a broader range of risk indicators, including direct damage and, in the case of Dottori et al. (2018), also indirect damage. In most studies where vulnerability has been considered, this has been done by using (one or a limited number of) intensity-damage

functions (IDFs), namely depth-damage functions, whilst Jongman et al. (2015) and Dottori et al. (2018) have also used vulnerability ratios. The only study to develop dynamic vulnerability scenarios is that of Jongman et al. (2015).

There has also been a clear shift from studies using non-probabilistic approaches focusing on impacts in a given year or time-period, or on one or several discrete return period, to approaches using a probabilistic approach. The latter studies have integrated impacts across a range of return periods to estimate risk in terms of expected annual impacts. To date, none of the

global scale studies use probabilistic stochastic event sets.



Planned future DRR measures have only been considered in the studies of Winsemius et al. (2016), Ward et al. (2017), and Willner et al. (2018), and only through structural measures. Of these studies, only Ward et al. (2017) assesses the associated costs and benefits. No studies to date have assessed behavioural aspects of future DRR measures. All studies project huge increases in future absolute risk, assuming no future DRR measures, on the order of hundreds to thousands of percent

depending on indicator, scenario, time-periods, and study. However, the increases are significantly lower when expressed relative to population and/or GDP. Moreover, the more recent papers that include either change in vulnerability or future DRR measures show that much of the projected future risk could be reduced if effective DRR measures are taken, often with the benefits of these exceeding the costs.

### 2.1.2. Coastal floods

As with river floods, a relatively large number of studies have been carried out to assess global risk from coastal floods. Details of each study can, therefore, be found in the Supplementary Information, and they are summarised in this section and in Table 1.

Pioneering work on global coastal flood risk was performed in the *Global Vulnerability Assessment* for the Intergovernmental Panel on Climate Change (IPCC) (Hoozemans et al., 1993). This study used empirical approaches to derive extreme sea levels

for four different return periods using co-variables such as wind and pressure climatologies and bathymetry. These were combined with socioeconomic data and scenarios, as well as cost estimates for increasing protection standards to estimate present day and future risk, as well as the costs of future DRR measures.

This method was extended in the European *DINAS-COAST* project, in which the Dynamic Interactive Vulnerability Assessment model (DIVA) was developed. This model assesses coastal flood risks at a finer spatial resolution than Hoozemans

(1993) using coastal segments, whereby each segment represents a different coastal archetype (Vafeidis et al. 2008). In total, 12,148 coastal segments are defined, with the length of each section ranging from less than 1 to 5,213 kilometres, depending on physical and socioeconomic conditions. For these segments, extreme sea levels were computed following Hoozemans et al. (1993), and combined with biophysical and socioeconomic data along these coastal segments to enable assessment of risk in terms of affected coastal area and population, population forced to migrate, as well as damage based on a single depth-

damage function. From the start, the DIVA model has explicitly represented the costs and benefits of future DRR measures, since it was developed to assist climate adaptation studies. DIVA has been used in several studies at global scale, including Hinkel and Klein (2009), Hinkel et al. (2010), and Hinkel et al. (2014) to assess risks under scenarios of sea level rise, subsidence, and population growth, with several assumptions on future DRR measures. It has also been used by Hallegatte et al. (2013) to assess coastal flood risk and the costs and benefits of structural DRR measures in major coastal cities by 2050.

Jongman et al. (2012) used extreme sea level estimates from the DIVA studies to produce a gridded inundation map for a 100 year flood. Using this, they assessed the increase in risk during the 21st century as a result of change in exposure only. More recently, Schuerch et al. (2018) modified DIVA to perform a more comprehensive assessment of coastal wetland responses to climate change globally. Combined, the aforementioned studies show that in general risk will increase by a large amount





throughout the 21st century if no future DRR measures take place. The costs of implementing future DRR measures are large,
but are far smaller than the benefits gained by their risk reducing effect.

The previous studies all use the extreme sea levels from the original DIVA model. Fang et al. (2014) present the first gridded
analysis of current flood risk in terms of affected people and affected GDP. Not until 2013 were the extreme sea levels in
coastal flood risk studies improved through hydrodynamic modelling. Muis et al. (2016) made the dynamic Global Tide and
Surge Reanalysis (GTSR) - the first of its kind - using the physically-based global coverage Global Tide and Storm surge
Model (GTSM). GTSR was validated against sea level observations world-wide and compared against the previously used
extreme sea levels within DIVA. GTSR represents extreme sea levels much better than the previously used extreme sea levels.
Muis et al. (2016, 2017) used these new extreme sea levels to estimate the population exposed to a 100 year return period
flood, and found the global numbers to be about 28% lower than those using the previously used extreme sea levels. Recent
work by Hunter et al. (2017) used the approach of Hallegatte et al. (2013) to assess flood risk in global cities, but using extreme
sea levels derived from tide gauges. They find that the original extreme sea levels are overestimated compared to observations,
and that average annual damages are about 30% lower using observations.

Several recent studies have examined new avenues for examining coastal risk. Beck et al. (2018) estimate the global flood
protection savings that can be provided by coral reefs. Vafeidis et al. (2019) investigate the uncertainty introduced to global
coastal risk modelling by the flood attenuation land inwards. They show that the uncertainties in attenuation are similar in size
to the uncertainties in sea level rise, suggesting this is an important future research direction.

From the outset, coastal flood risk assessments have been forward-looking, since the original studies were developed to assist
climate adaptation studies. The use of future hazard and exposure scenarios is therefore widespread, as is the assessment of
the costs and benefits of DRR measures, both structural and nature-based. Vulnerability has been included in many of the
DIVA studies, dynamic vulnerability scenarios have not been developed. Several recent studies have focused more on current
risk, using the newer datasets on extreme sea levels. Recently, a number of studies have assessed risks at a 30" x 30" gridded
resolution, moving towards a higher spatial resolution compared to the studies carried out using the DIVA model. Coastal
flood risk studies have either assessed impacts for a single return period, or have assessed probabilistic risk by aggregating
across several return periods. As with river floods, to date none of the global scale studies use probabilistic stochastic event
sets.

### 2.1.3. Pluvial floods

To the best of our knowledge, the scientific literature does not contain any examples of global scale pluvial flood risk
assessments, i.e. flooding caused by intense rainfall that exceeds the capacity of the drainage system. Pluvial flooding is most
commonly assessed using flood models for a small area (e.g. city or even part of a city) that generate data on depth and velocity
of surface water associated with rainfall events of different intensities. Guerreiro et al. (2017) did develop a modelling approach
to assess pluvial flood hazard for 571 cities at the continental scale in Europe. The paper outlines some of the key challenges
in such a continental approach, which would be amplified for a potential global scale application. These include: difficulties



in obtaining the required hourly rainfall records; low resolution of continental to global Digital Elevation Models (DEMs) compared to those typically used for pluvial flood models; and the lack of data to represent local sewer systems, building shapes, and infiltration in local green spaces. Opportunities to collect such data lie in high resolution remote sensing and data

science (Schumann and Bates, 2018), but also in local crowd sourced methods such as community mapping (Winsemius et al., 2019), which is a promising avenue particularly in strongly growing urban centres in developing countries. Guerreiro et al. (2017) demonstrate that current modelling capabilities and requisite computing power make large scale pluvial flood hazard assessment a possibility, if these data challenges can be overcome.

### 2.2.  Tropical cyclones

Several studies on tropical cyclone (TC) risk have been carried out at the global scale, using various methods (Table 1). Peduzzi et al. (2009) assess current TC risk hotspots, expressed in terms of expected number of fatalities per year per country. Hazard is represented by computing buffers along individual TC tracks between 1980-2000, where wind speed exceeds a threshold of 42.5 m/s. The tracks are taken from the the PreView Global Cyclones Asymmetric Windspeed Profile dataset. The average cyclone frequency per cell is determined by taking the spatial extents of individual cyclones (5km x 5km), and averaging the

frequency over the entire period. Exposure is represented by population and GDP per capita (5km x 5km); taken from GRID and the World Bank, respectively. Vulnerability is represented using a selection of 32 socioeconomic and environmental variables. Since the results are only for current conditions, future DRR measures are not accounted for.

Cardona et al. (2014) also assess current risk from TCs in the current time-period. They express risk in terms of economic damage (average annual loss and probable maximum loss for fixed return period of 250 years) by combining data on hazard,

exposure, and vulnerability within the *CAPRA Platform Risk Calculator* (www.ecapra.org). Hazard is represented by maps of wind speed for different return periods at a horizontal resolution of 1km x 1km. These are derived from a stochastic set of wind fields, calculated using a model forced by historical observations of TCs from the IBTrACS v02r01 dataset (Knapp et al., 2010). Exposure is represented using the common dataset of the GAR 2015, in which it is represented as a group of buildings in each point or cell of analysis with a resolution of 5km x 5km. Vulnerability is represented by IDFs relating

maximum wind velocity sustained for five seconds gusts at ten metres above ground level (Yamin et al., 2014). Since the results are only for current conditions, future DRR measures are not included. In the paper, rankings of national level risk are shown.

Fang et al (2015) assess the global population and GDP at risk from TC winds in the current period. To represent hazard, they develop a 6-hourly TC track database up to 2012 using CMA-track (Ying et al 2014), HURDAT, (Landsea and Franklin 2013)

and IBTrACS (Knapp et al., 2010). To convert the gust wind to sustained wind speed, a gust factor model is applied. Next, a Gumbel distribution is fitted to all grid cells (30" x 30") with more than 20 TC events to calculate wind speeds at several return periods. Exposure is represented using gridded population data at a resolution of 30" x 30" from LandScan 2010 from ORNL, and is represented using gridded GDP data at a resolution of 0.5 °x 0.5° from the Greenhouse Gas Initiative (GGI) dataset of IIASA for 2010. Vulnerability is not accounted for, and since the study only examines current risk, no future DRR measures



are included. They find that China has the highest expected annual affected population and GDP, and the top 10% of countries are largely in Asia.

Peduzzi et al. (2012) assess risk from TCs over the period 1970-2009, with projections of future risk to 2030 based on projections of increased exposure only. Hazard is represented by maps showing TC frequency and maximum intensity for events between 1970 and 2009 at a horizontal resolution of 2km x 2km. The dataset is derived from a TC model of wind speed
profiles using a parametric Holland model (Holland, 1980), which is forced using historical observations of TCs from the IBTrACS v02r01 dataset (Knapp et al., 2010). Exposure is represented by gridded maps of population and GDP at a horizontal resolution of 30" x 30". Current population data are taken from LandScan 2008 (LandScan, 2008) and current GDP data are taken from data from the World Bank. They are both extrapolated to each decade from 1970 to 2030, based on UNEP country data. Vulnerability is represented using different country level parameters relating to the economy, demography, environment,
development, early warning, governance, health, education, and remoteness. These are used to calculate exposed GDP, affected population, and mortality risk per country for each year between 1970-2030. Future DRR measures are not included. The average population exposed to TCs per year is projected to increase by 11.7% by 2030, with about 90% of this increase in Asia. In relative terms, the largest increase in risk is in Africa.

Mendelsohn et al. (2012) assess TC risk using future scenarios of hazard, exposure, and vulnerability.  Risk is expressed in
terms of direct damage, and the damage pertains to storm surge, wind, and freshwater flooding, without distinguishing between the three sources. Hazard is represented by TC landfall locations and intensity from a synthetic dataset of TC tracks, simulated using the model of Emanuel et al. (2008). The TC model is seeded with climate data from four GCMs: CNRM-CM3, ECHAM-5, GFDL-CM2.0, and MIROC-3.2, for both the current period (1981-2000) and future period (2081-2100) under the SRES A1b emissions scenario; sea level rise is not accounted for. The damage per storm is calculated using a statistical damage
function per county in the USA or per country for the rest of the world. The damage function uses population density to represent exposure, and income as an indicator of vulnerability. Current exposure and income data are used per county for the USA and per country for other regions. Future projections of population per country are taken from the World Bank (2010), and future projections of GDP per country are based on long-term growth rates for three income groups. Probabilistic risk is expressed as direct damage per year, based on the damages from the stochastic storm tracks. Future DRR measures are not
included. The main findings are an increase in global damage of ~115% by 2100 due to changes in population and income, with approximately a further doubling due to climate change.

In summary, several studies have examined the risk from TCs at the global scale. Most have only considered current conditions, except Peduzzi et al. (2009) (change in exposure) and Mendelsohn et al. (2012) (changes in hazard, exposure, vulnerability). A defining aspect of a TC hazard is that it is composed of wind, precipitation, and storm surge. However, the current studies
to date do not explicitly model all of these aspects. At present there are no methods available to parametrically model 2D precipitation fields from TCs in the same way that wind fields are parametrically modelled using the Holland Model (Holland 1980). A large range of different approaches have been used for defining and modelling both the hazard and risk. Mendelsohn





et al. (2012) and Cardona et al. (2014) express risk in probabilistic terms. Mendelsohn et al (2012) is the only study to use a synthetic TC dataset, whilst the other studies use historical TC events to construct the hazard.

## 2.3. Hazards associated with severe convective storms

Perils associated with severe convective storms (SCS), such as large hail, heavy rainfall, strong wind gusts, or tornadoes, are among the most important perils in several regions of the world (Christian et al., 2003; Virts et al., 2013; Cecil et al., 2014). Of all SCS-related hazards, hail causes the largest economic damage (Kunz and Geissbuehler, 2017). However, the modelling of SCS risk is still in its infancy (Allen et al, 2016; Martius et al., 2018). As a result, there are currently no global risk models available for these SCS events or their sub-perils (hail, wind gusts, tornadoes, heavy rain, lightning), nor are efforts being made in this direction.

However, as insurance companies have to either provide the solvency capital for rare events (e.g. 200-year return period event according to the EU Solvency II directive in Europe) or have to reinsure their risk, there is a growing and large demand to better estimate the risk related to the different SCS sub-perils in a scientific manner (Allen et al., 2019). The few existing models owned by the insurance market quantify the risk mainly on a regional (e.g., Schmidberger, 2018a), national, or continental level (e.g., Punge et al., 2014). These models are not freely available and the literature is scarce. Initial loss estimates in the insurance industry were based on mathematical analyses of the company's own damage data and portfolio. As more and more data on SCS events and their sub-perils have become available during the last decade, an increasing number of damage and risk models for SCS have been developed either by insurance companies or by companies developing catastrophe models (CAT models). The CAT models consider basic hazard characteristics (e.g. length, width, angle of the footprints of sub-perils) and intensity metrics (e.g. hailstone size, hail kinetic energy, precipitation amount, max. wind speed) of an underlying event set and quantify the damage for a certain portfolio (exposure data) via fragility curves, a kind of IDF. Due to the restricted record length of SCS hazard event sets, stochastic simulations based on, for example, statistical distributions of the relevant input parameters in combination with Markov or Poisson processes are performed (e.g. Punge et al., 2014; Holmes, 2015; Ritz, 2017; Schmidberger, 2018a,b). In some cases, the CAT models also consider atmospheric conditions relevant for SCS development. None of the insurance models consider projected changes in the frequency and severity of SCS due to climate change.

## 2.4. Droughts

A relatively large number of studies have been carried out to assess global risk from droughts. Therefore, these are summarised in this section and in Table 1, and further elaborated upon in Supplementary Information. Here, we focus specifically on drought risk, rather than water scarcity. Drought itself is a difficult concept to define, and as a result more than 150 indices have been developed for its identification over the past decades. Commonly, drought as a hazard us defined as a relative concept, mostly as a deficit to normal e.g. when such drought indices fall below/exceed a given threshold. Nevertheless, since drought is a complex hazard which propagates from a rainfall deficit (meteorological drought), to soil moisture drought to



hydrological drought, variety of corresponding impacts may results with regard to the different types of drought, as well as underlying socio- economic and ecological conditions. Hence, since a universal of drought seems impracticable, a paradigm shift towards a definition of drought by its impacts has been recommended (Lloyd-Hughes 2014).

The multifaceted aspects of drought are reflected in the large range of risk indicators used in the studies described below, as well as the very diverse range of approaches and datasets used to represent hazard, exposure, and vulnerability. The studies

described below neither explicitly include future DRR measures, nor assess risk in a probabilistic sense.

Dilley et al. (2005) is one of the first to conduct a global scale assessment of drought risk, overlaying layers of hazard (Weighted Anomaly of Standardized Precipitation; WASP) and exposure (population, GDP, road density) information at a relatively coarse resolution of 2.5° x 2.5° for the current time period. They assess risk in terms of affected GDP, population, roads, and infrastructure. Christenson et al. (2014) build on the drought risk assessment of Dilley et al. (2005), by making a further

distinction between the type of population exposed (i.e. urban or rural). Neither of the aforementioned studies assess vulnerability. The studies of Yin et al. (2014) and Carrão et al. (2016) do include vulnerability (as well as hazard and exposure), and also calculate risk at a higher spatial resolution of 0.5° x 0.5°. Yin et al. (2014) express risk in terms of maize yield, and represent vulnerability by fitted logistic regressions between historical maize crop loss estimates and simulations of drought stress. Drought hazard is represented as the normalised cumulative water stress index during the growing season, and exposure

is represented by a map of fields and maize yield. Carrão et al. (2016) assess risk using a drought index integrating several factors. Vulnerability is represented in the form of proxies of economic, social, and infrastructural vulnerability, at resolutions from 5' x 5' to country scale. Hazard is represented by WASP, whereby a drought is identified when the monthly precipitation deficit is less than or equal to 50% of its long-term median value for three or more consecutive months. Exposure is represented by gridded maps of agricultural land, population, and livestock. On the basis of their results, Carrão et al. (2016) state that a

reduction in drought risk could be rapidly achieved by improved irrigation and water harvesting in regions where infrastructural vulnerability is high.

Several studies assess future drought risk, as a result of either hazard, or hazard and exposure. The forward-looking studies below all use a horizontal resolution of 0.5° x 0.5°, except for that of Smirnov et al. (2016), where a resolution of 2° x 2° is used. The majority of these studies do not include vulnerability (Arnell et al., 2013; Smirnov et al., 2016; Arnell et al., 2018;

Liu et al., 2018), whilst Li et al. (2009) and Guo et al. (2016) do include vulnerability, but only under current conditions.

Li et al. (2009), Arnell et al. (2013), and Guo et al. (2016) perform future simulations by projecting changes in hazard only for different time slices up to the end of the 21st century, as a result of climate change. As with the current drought risk studies, the risk, hazard, exposure, and vulnerability are represented using very diverse metrics. All assess risk in terms of agricultural impacts, although the metric used is different in all cases. Hazard is represented by the Palmer Drought Severity Index (PDSI),

Standardised Precipitation Index (SPI), namely SPI-12, and normalised cumulative water stress index respectively, whereby different thresholds are used to identify hazardous drought conditions. Representation of exposure is diverse, but shows some similarities across the studies, being represented either by data on crop yields, cropland/field areas, or a combination of both.



Vulnerability is included in very different ways; the proportion of area equipped for irrigation per country in Li et al. (2009) and proxies of economic, social and infrastructural vulnerability in Guo et al. (2016).

Smirnov et al. (2016), Arnell et al. (2018), and Liu et al. (2018) add a layer of complexity in their future drought risk assessments by including projections of change in exposure as well as hazard, although vulnerability is not included in these studies. Risk is expressed in terms of the population affected in all of the studies, as well as cropland area affected by Arnell et al. (2018). Again, the indicators used to represent hazard are very diverse: Standardised Precipitation Evapotranspiration Index (SPEI), namely SPEI-24; Standardised Runoff Index (SRI); and PDSI respectively. Since they all assess the population

affected by drought, they all use gridded population projections for the current and future time period, with Arnell et al. (2018) additionally using crop data.

As a result of the wide range of risk indicators and approaches used, estimates of global drought risk vary significantly from study to study. Nevertheless, all studies find a robust increase in future drought risk due to changes in both climate and socioeconomic conditions.

**2.5.  Wildfire**

Wildfire is increasingly understood as being an inherently socio-natural phenomenon with feedbacks between society, vegetation, fire weather, and climate (Riley et al., 2019). However, global wildfire risk is a particularly understudied area of disaster risk assessment. This may be due to a focus on global burnt area products to provide input to global climate modelling as a significant source of emissions (Giglio et al., 2009; GCOS, 2011; Chuvieco et al., 2016) rather than a disaster risk

emphasis. It is also due to the large degree of complexity with interactions between natural and human processes driving occurrence and intensity of wildfires as well as exposure and vulnerability to them.

In the studies reviewed below, DRR measures are not explicitly accounted for in the modelling frameworks. This is largely due to uncertainty in human actions - continued ability to manage fuel and suppress fires under different climatic conditions and increased sprawl into wildland-urban-interface (WUI) areas; and on regime changes in weather and vegetation making

previously non-hazardous vegetation areas susceptible to fire, especially with increased logging and fragmentation, particularly in tropical areas (Laurance and Williamson, 2001; Flannigan et al., 2009; Corlett, 2011; Jolly et al., 2015). Significant steps forward could be made by examining interactions between vegetation, weather, climate, and human activities that cause wildfires.

Meng et al. (2015) map forest wildfire risk globally under current conditions only. Risk is expressed in terms of forest area

burnt for different return periods at a resolution of 0.1° x 0.1°, but are not integrated to estimates of probabilistic risk. Hazard is represented as annual forest wildfire occurrence based on historical data from MODIS satellite imagery (0.1° x 0.1°). The short historical time series of wildfire occurrence (12 years) required the use of fuzzy mathematics to allow for the calculation of different return periods of forest wildfire (Huang, 1997; Huang, 2012). Exposure is represented by the area of forest (representing economic value), using land cover data at 0.1° x 0.1°. Vulnerability is modelled as a function derived from fire

occurrence versus burnt area, whereby a cell with high vulnerability indicates that a small number of fires can cause a large



amount of burnt area, with the inverse being true for low vulnerability. The results show highest risk in central Africa, central South America, north-western Southeast Asia, mid-eastern Siberia, and the northern regions of North America. High risk can also be seen eastern Australia, Laos, Cambodia, Thailand, Bangladesh, eastern Russia, and the borders of Zambia, Angola and Democratic Republic of Congo. Risk results from Meng et al. (2015) are used in Shi et al. (2015) as part of a multi-hazard

global risk assessment. In this study, impacts are integrated across return periods to estimate probabilistic risk. However, the results are aggregated across eleven different hazards, so the risk attributed to wildland fire cannot be identified.

Cao et al. (2015) is a similar study to Meng et al. (2015), but presents results for grassland wildfire risk at a resolution of 1km x 1km. The analysis is carried out for current conditions, expressing risk as average impacts per year over 2000-2010. Hazard is calculated based on probability of ignition, slope and vegetation properties (calculated from MODIS data) (1km x 1km). A

logistic regression model is developed using these properties and historical burnt area records to model the probability of grassland burning. Exposure is based on the assumption that the primary impact of grassland fires is on the stock industry, which is dependent on available biomass from grassland. Therefore global net primary product (NPP) (1km x 1km) is used as the proxy for exposed value for grassland wildfire. Vulnerability is the probability of fire spread or propagation and contributes to the calculation of probability of grassland burning. The key results show that the areas of highest risk are in Australia, Brazil,

Mozambique, Madagascar, United States of America, Russia, Kazakhstan, China, Tanzania, Canada, Angola, South Africa, Venezuela, Argentina, Nigeria, Sudan and Colombia. Again, the results are used in Shi et al. (2015) as part of a multi-hazard global risk assessment.

Knorr et al. (2016) present wildfire risk for 1901 to 2005 and then to 2100 using Representative Concentration Pathways (RCPs) and Shared Socioeconomic Pathways (SSPs) to project hazard and exposure. Risk is expressed in terms of affected

people and burnt area per year at a resolution of 1° x 1°. Hazard is modelled by coupling a semi-empirical fire model, SIMFIRE (Knorr et al., 2014) with a global dynamic land ecosystem and biogeochemical model LPJ-GUESS (Ahlström et al., 2012). This produces hazard metrics of fractional burnt area per year. The coupling with LPJ-GUESS allows vegetation and weather factors to update, with SIMFIRE providing annual updates on fire frequency per grid cell. The hazard modelling is driven using RCP4.5 and 8.5 data for monthly mean precipitation, temperature and radiation from CMIP5. Exposure is represented

by population under SSP 2, 3 and 5, although this does not include changing rates of population splits between urban and rural areas, which is an important factor for future interactions with wildfire. Vulnerability and DRR measures are not explicitly considered, although population density is a factor within SIMFIRE accounting for the idea that generally increasing population will suppress wildfires. Under the future scenarios, there is a significant increase in the number of people in fire prone areas between 1971-2000 and 2071-2100: between 23% and 56% for RCP 4.5, and 25% and 73% for RCP 8.5, averaged

across SSPs.

### 2.6. Earthquakes

A relatively large number of studies have been carried out to assess global risk from earthquakes. Therefore, these are summarised in this section and in Table 1, and elaborated on in Supplementary Information. Several global assessments have





used index-based methods based on overlays to assess global earthquake risk, and are not explicitly discussed in this review.
All of the studies reviewed in Table 1 explicitly include hazard, exposure, and vulnerability, although none of them carry out future projections of risk.

One of the first global earthquake risk models to go beyond this approach is that of Chan et al. (1998), which examines direct damage via macroeconomic indicators to derive global seismic loss at the relatively coarse scale of 0.5° x 0.5°. The study of Dilley et al. (2005) also uses hazard data based on past events (based on the Richter scale), but in addition uses 50 year return
period hazard maps from the Global Seismic Hazard Program (GSHAP). The resolution of the risk analysis is higher, at 2.5' x 2.5', and a wider range of impact indicators is used (affected population, affected GDP, affected road and rail infrastructure, and fatalities). Global studies at a higher resolution of 1km x 1km were performed by Jaiswal and Wald (2010, 2011). These two studies essentially use the same approach, but the former assesses risk in terms of affected people and fatalities, whilst the latter also assesses risk in terms of direct economic damage and affected GDP.  Hazard is represented by shakemaps of intensity
from past events at a resolution of 1km x 1km.

Daniell (2014) and Daniell and Wenzel (2014) also assess global earthquake risk at 1km x 1km, whereby risk is expressed in terms of direct and indirect damage, fatalities, affected people, and affected GDP, at a resolution of 1km x 1km. In this case, hazard is represented by the spectral acceleration and/or macroseismic intensity (MMI) at each point, which is then rasterised on a 1km x 1km grid for each event. None of the aforementioned studies assess risk probabilistically, instead calculating
impacts for past events or for a given return period.

Li et al. (2015) describe a global earthquake risk model using a probabilistic approach, in which the impacts are integrated over several exceedance probabilities. Hazard is represented by peak ground acceleration (PGA) at 0.1° x 0.1° with conversion to macroseismic intensity. Risk is expressed in terms of direct damage, fatalities, affected people, and affected GDP, at a resolution of 0.5° x 0.5° for mortality and 0.1° x 0.1° for economic-social wealth.
From the GAR2013 onwards (UNDRR, 2013, 2015a, 2017), a probabilistic approach using stochastic hazard modelling has been used in the GARs. Risk, in terms of direct damages, is calculated stochastically at a country resolution; and expressed at national scale in terms of Probable Maximum Loss and Annual Average Losses. Hazard is represented by spectral accelerations at a resolution of 5km x 5km, using a stochastic event set of earthquakes around the world. The Global Earthquake Model (GEM) (Silva et al., 2018) also uses stochastic event sets to produce probabilistic risk estimates in terms of economic damage
to buildings (1km x 1km).

Exposure data used in global earthquake models has generally increased in resolution from ~0.5° to ~1km, usually using datasets such as gridded population and GDP. A defining feature of more recent global studies has been the use of capital stock data (e.g. Daniell, 2014; Daniell and Wenzel, 2014; Silva et al., 2018). Vulnerability is represented in various ways, ranging from empirical loss and fatality functions based on reported losses and fatalities, empirical fatality and loss ratios; and IDFs
based on different building types.





### 2.7. Tsunamis

As with other natural hazards, tsunami risk can be broken down into hazard, vulnerability and exposure. However, limited empirical data on vulnerability and the computational burden of tsunami wave propagation simulations means that only a limited number of studies have so far been developed, and most of them either have limited scope or resolution. Early risk assessments at local scale from Berryman et al. (2005) and Grezio et al. (2012) use methods to estimate the inundation at low resolution. The highest spatial resolution simulations have been carried out by Wiebe and Cox (2014) for parts of Oregon, Jelinek et al (2012) for Cadiz in Spain, and De Risi and Goda (2017) for the Miyagi prefecture in Japan. Some models have assessed risk in terms of fatalities, such as Tinti et al. (2008) or Okumura et al. (2017).

Løvholt et al. (2015) assess global risk in terms of direct damage, affected people, and fatalities at a resolution of 1km x 1km. Risk is calculated probabilistically using stochastic event sets and a Probable Maximum Loss curve at 100, 500 and 1500 years, as well as average annual losses. Risk is only assessed for current conditions, and therefore no DRR measures are included. Hazard is represented in terms of inundation depth at a resolution of 90m x 90m. This is simulated using a 25km propagation model, followed by a 2D-model at a resolution of 1' x 1', and then further downscaled using a 90m DEM. Exposure is represented by capital stock and population at 1km x 1km, taken from the coastal database of De Bono and Chatenoux (2015) used as part of GAR2015. Vulnerability is represented by IDFs for different buildings types derived from expert workshops (Maqsood et al., 2014) for the Asia Pacific. The Suppasri et al. (2013) functions are used for the rest of the world. They find that Japan and the Philippines, among other island nations, have the highest relative risk.

Schäfer (2018) introduce a generic tsunami risk assessment framework that could be applied for global risk modelling, and apply it to various regions around the world. Risk is expressed in terms of direct damage, affected GDP, affected population, and fatalities, for current conditions, not accounting for DRR measures. It is calculated at 90m x 90m resolution for various return periods, with expected annual impacts being assessed by integrating across different exceedance probabilities. Hazard is represented by maps of inundation depths (90m x 90m), based on numerical simulations using shallow water wave equations and machine learning. Exposure is derived using a population-based capital stock model (90m x 90m) built from the Global Human Settlement Layer (Pesaresi et al., 2016). Vulnerability is represented by depth-damage IDFs that are resolved for three building classes (light, moderate and massive buildings) and a fatality function considering both water depth and arrival time. The study is tested in various regions, including Japan, Chile and the Caribbean and derives average annual losses and probable maximum loss curves.

### 2.8. Volcanoes

Volcanoes can produce a variety of hazards, including pyroclastic density currents (pyroclastic flows, surges and blasts), tsunamis, lahars, tephra (including volcanic ash and ballistics), debris avalanches (sector collapse), gases and aerosols, lava flows and domes and lightning. Not all volcanic hazards are produced by every volcano or eruption, and some can occur without an eruption, for instance volcanic gas. Whilst many hazards might be triggered by the volcano directly, the occurrence



or distribution of others can be influenced by hydrometeorological factors, for instance, rainfall triggered lahars and landslides or the influence of wind on the distribution of volcanic ash. Comprehensive assessments of volcanic hazard or risk at the global

scale do not exist. Even assessments at the local scale are unlikely to include the effect of DRR measures. The field of volcanic hazard and risk assessment can therefore appear less well developed compared to other natural hazards, with the effect that assessments combining multiple natural hazards typically underestimate the threat from volcanic activity. There are at least three key factors that limit our ability to assess volcanic risk at the global scale: the multi-hazard, time-varying, and complex nature of volcanic events; a large discrepancy in the quality and quantity of data required to inform global volcanic hazard

assessments between regions; and limitations of data to inform global volcanic risk assessment, especially because large, damaging eruptions impacting populated areas are relatively infrequent and impacted zones can be dangerous and sometimes inaccessible for long periods. International collaborations have now been established to facilitate the production of systematic evidence, data, and analysis of volcanic hazards and risk from local to global scales (e.g. Loughlin et al., 2015; Newhall et al., 2017; Bonadonna et al., 2018).

Past approaches to global volcano risk, described below, have mainly aimed to identify those volcanoes or cells that pose the greatest relative danger, in order to inform subsequent in depth investigations using local data and knowledge. The first assessment of the hazard threat posed by volcanoes globally is Yokoyama et al. (1984). They use an index-based approach to identify 'high-risk' volcanoes. Binary indices are used to score ten hazard and five exposure components, describing the frequency of recent explosive activity and hazards, and the size of the population within a certain radius of the volcano,

respectively. Two quasi-vulnerability binary scores are used: one for if the volcano had produced historical fatalities, and one for if evacuations had resulted from historical eruptions. Scores for each volcano are summed, with 'high-risk' volcanoes defined as those with a score ≥ 10. This approach results in notable volcanoes that went on to produce some of the worst volcanic catastrophes of the 20th century being considered *not* high-risk.

Small and Naumann (2001) and Freire et al. (2019) rank volcanoes globally according to the population exposed within radii

of 200 km and 10-100 km from a volcano, respectively. Hence, hazard and vulnerability are not included. Exposure is represented by gridded population data at a resolution of 2.5' x 2.5' (Small and Naumann, 2001) and 250 m (Freire et al., 2019). For the GAR2015, an index-based approach to assessing volcanic hazard and risk is also used, combining data and the approaches of Auker et al. (2015) and Brown et al. (2015a,b). Hazard is represented by Auker et al. (2015) using an index method to represent the hazard level of each volcano according to the frequency and intensity of past eruptions, while

accounting for record incompleteness. Exposure is represented using the method of Brown et al. (2015a), an index of population exposure weighted to historical data of fatalities (Auker et al. 2013) with distance, and vulnerability is expressed in terms of affected population and fatalities for past events. Combining the hazard index with fatality-weighted population counts gives estimates of individual volcanic threat. Not all volcanic hazards are considered in the weighting; the weights are sourced from volcanic flow fatality data only, as pyroclastic density currents and lahars are the source of the majority of direct

historical fatalities (Auker et al. 2013). Brown et al. (2015b) further aggregate the volcanoes to the national scale to provide



country-level estimates of volcanic threat. Whilst this approach represents our most sophisticated attempt at considering volcanic hazard and risk at the global scale, it still assumes concentric radii around each volcano.

Grid-based global assessments of volcanic risk have been carried out by Dilley et al. (2005) and Pan et al. (2015). Dilley et al. (2005) include volcanoes as part of their multi-hazard hotspot analysis, whereby risk is expressed in terms of affected GDP
and population, and fatalities (2.5' x 2.5'). Hazard is expressed in terms of the count of volcanic activity, which is gridded to 2.5° x 2.5°, between 79-2000 A.D. (i.e. no consideration of the size of eruptions or their spatial extent). Exposure is represented by gridded population from the Gridded Population of the World 3 (GPWv3) dataset from CIESIN; GDP per capita at national scale from the World Bank; and transportation lengths from the VMAP datasets, all at 2.5' x 2.5'. Vulnerability is represented by two quasi-vulnerability values, for mortality and economic loss rates globally as a result of volcanic activity between 1981
and 2000 based on the EM-DAT database, aggregated to country and regional levels. A major limitation of this approach, as recognised by the authors themselves, is that the hazard records are too short to capture the larger, typically more damaging events and are biased towards those volcanoes for which we have good and recent records of past activity. Also, no attempt was made to account for far-reaching volcanic hazards like ash. Pan et al. (2015) built on this approach to assess fatalities, by extending the length of the fatalities database used to 1600 AD. They also add relationships between eruption Volcanic
Explosivity Index (VEI) and frequency and hazard extent, although concentric circles are still assumed. Hazard is defined here as the frequency of each VEI eruption, using the method of Jenkins et al. (2012a). Exposure is represented using gridded population data (30" × 30") of 2010 from Oak Ridge National Laboratory (ORNL) (Bright et al 2011). Vulnerability is represented by fatality curves fitted to the historical average fatality of each VEI provided by National Oceanic and Atmospheric Administration (NOAA).

**2.9. Landslides**

Assessing the risk associated with landslides at a global scale is challenging for several reasons. Firstly, the spatial extent of individual landslides is typically small, limiting the effectiveness of routinely monitoring these events at global scale. Further, the diversity of parameters influencing the hazard susceptibility (e.g. elevation, lithology), preconditioning (e.g. soil moisture, seismicity), and triggering (e.g. extreme rainfall, earthquakes) make it difficult to physically model these processes using
uniform approaches. Moreover, while the spatial extent of landslide source regions is generally small, the downstream hazards - which are often associated with the greatest damage (Badoux et al., 2014) - can be more widely distributed, meaning risk assessments must consider locations both near and far from the source areas.

Efforts have been made to catalogue landslides and their impacts, based on a range of data sources including media reports, government statistics and other written sources, remote sensing, and citizen science (e.g. Guzzetti et al. 1994; Kirschbaum et
al. 2010, Petley, 2012; Tanyas et al., 2017; Froude and Petley, 2018; Juang et al. 2019). In addition, several studies have tried to model global landslide hazard (e.g. Stanley and Kirschbaum, 2017; Kirschbaum and Stanley, 2018). In the following paragraphs, those studies that have explicitly assessed global risk are described, and summarised in Table 1.

Nadim et al. (2004, and updated in 2006) is among the first studies to assess risk associated with landslides at the global scale. Risk is expressed in terms of the number of fatalities per year at a resolution of 30" x 30", over the period 1980-2000, associated





with landslides and avalanches. Hazard is estimated using a range of global datasets based on both susceptibility (factors such as slope, lithology, and soil moisture) and triggering factors (rainfall and seismicity). A similar approach is used to define the hazard associated with avalanches. Exposure is represented by the Global Population of the World v4 (GPWv4) dataset (30" x 30") (CIESIN 2016). Vulnerability is estimated using empirical data on loss-of-life from landslides in a number of countries, using the EM-DAT database. Future projections and therefore future DRR measures are not included in the study. They find
that hotspots for landslide fatalities are the Himalayas, Taiwan, the Philippines, Central America, northwestern South America, the Caucasus, Indonesia, Italy, and Japan, with smaller proportional impacts in other countries and regions.

Dilley et al. (2005) estimate the risk associated with landslides, expressed in terms of direct damage and affected GDP and population at 2.5' x 2.5', for current conditions. Hazard is represented using the data of Nadim et al. (2004) at 30" x 30". Exposure is represented by data at 2.5' x 2.5' constructed from GPW population data, road density data from VMAP datasets
developed by the United States National Imagery and Mapping Agency, and gridded economic and agricultural activity from the World Bank and based on Sachs et al. (2001). Vulnerability is represented by empirical loss rates based on the EM-DAT database. Future projections and therefore future DRR measures are not included in the study. Their primary output is in map form, with the most elevated impacts found in many of the same locations as Nadim et al. (2004, 2006), although they find higher impacts in terms of total GDP in China.

Yang et al. (2015) assess risk in terms of fatalities for the current period at a resolution of 0.25° x 0.25°. Hazard is represented based on the method of Nadim et al. (2006), using TRMM satellite rainfall data to estimate the number of landslide events, and filling in gaps in the data using information diffusion theory. Exposure is represented by population data from Landscan, resampled to 0.25° x 0.25°. Vulnerability is based on empirical mortality rates per country calibrated using a dataset of global landslides causing fatalities from Kirschbaum et al. (2010). Future projections and therefore future DRR measures are not
included in the study. They find similar patterns to Dilley et al. (2005) and Nadim et al. (2006), but additionally find many areas that have elevated risk of mortality compared to those prior studies, including large parts of China and sub-Saharan Africa.

Nowicki Jessee et al. (2018) present a global earthquake-induced landslide hazard model, which is implemented within the USGS Ground Failure hazard and risk model. Risk is expressed in terms of exposed population in near real-time for each
earthquake event that triggers landsliding. Hazard is calculated by leveraging the earthquake-triggered database from Tanyas et al. (2017), based on various sources of information describing factors controlling susceptibility (such as slope and lithology) and earthquake parameters, such as shaking intensity, to estimate the relative density of landsliding within an area impacted by a major earthquake. The model landslide density estimates are not dependent on resolution. Exposure is represented by Landscan population maps, with a resolution of 30" x 30" (Bright et al. 2017). No vulnerability data are incorporated. Future
projections and therefore future DRR measures are not included in the study. For each earthquake, an estimate of population exposed to landsliding and liquefaction is presented.



## 3. Comparison of approaches across hazard types

### 3.1. (Dynamics of) risk elements

As our review focuses on global scale natural hazard risk assessments, we have not included studies that only examine the
hazard. All but two of the studies have an explicit representation of hazard intensity and/or probability, and of exposure. The only exceptions are the volcano studies of Small and Naumann (2001) and Freire et al. (2019), in which the population living within a set radius of volcanoes is estimated, without an explicit representation of the hazard. About two-thirds of the reviewed studies include a specific representation of vulnerability. Across the various hazards, there is no clear difference in the proportion of studies including vulnerability as we move towards the most recent publications. It is noteworthy that all of the
earthquake and tsunami studies reviewed include a specific representation of vulnerability. For pluvial flooding and SCS, there are currently no global scale risk models, with the local scale of the hazard and impact of these events making their large scale modelling difficult.

In terms of the inclusion of dynamic risk drivers, there is a clear difference between studies focusing on hydrological, climatological, and meteorological hazards, and those focusing on geological hazards. For geological hazards, none of the
reviewed studies include future projections, whilst for hydrological, climatological, and meteorological hazards, around two-thirds of the studies include projections of at least one of the risk drivers. The difference between the studies of hydrological, climatological, and meteorological hazards, compared to those of geological hazards in terms of projections, may be due to the climate change-related focus of many studies in the former group. This provides a policy context for carrying out forward-looking hazard projections to examine the influence of climate change on risk. In total, there are 32 reviewed studies that
include future projections: 19 include projections of hazard and exposure; 8 include projections of hazard only; and 3 include projections of exposure only. The remaining 2 studies include projections of vulnerability (as well as hazard and exposure); 1 study for river flooding and 1 for TCs. Time-horizons used for the forward-looking studies tend to be towards mid and late 21st Century. For river flooding and drought, several recent studies have examined future warming levels, rather than future time-slices. It is interesting to note that projections of the other risk drivers have not yet been examined for the geological
hazards, despite their importance for other policy contexts, such as the Sendai Framework and SDGs, although definitions used for monitoring of these frameworks explicitly state disaster risk assessment as "evaluating of existing conditions of exposure and vulnerability" without considering future change (UNGA, 2016). The global scale geological risk studies could apply some of the forward-looking models of exposure in their analysis to estimate future risk, which is of importance for designing prospective risk management strategies and measures.

### 3.2. Resolution and type of input data

The list of studies in Table 1 shows a general tendency towards higher resolution risk analyses as we move towards more recent studies, though this is not the case for all studies. Many of the early studies are at resolutions from several degrees to 0.5°, and for many hazards there has been a progression towards higher resolutions of 30"/1km, and even up to 90m/point


values in some cases. We do see differences between the various hazards. Most of the early coastal flood risk studies examine
risk using the coastal segments from the DIVA database, although recent studies have also moved towards 30"/1km. For
drought, the resolution tends to be much lower, from around 0.5° to 2.0° in general. This is in line with the difference in
resolution at which drought impacts are felt (Stahl and Hisdal, 2004), compared to many of the other hazards in which the
direct impacts are felt more locally. For a similar reason, TC studies tend to show a much higher resolution in their analysis,
which reflects the fact that TC impacts for an individual event are felt over smaller areas. Most global volcano risk studies to
date examine risk at the resolution of an individual volcano, rather than on a raster grid. Several recent studies on earthquakes
(Silva et al., 2018) and earthquake-induced landslides (Nowicki Jessee et al., 2017) use model frameworks that can use
variables scales.

In most cases, the resolution of the risk analysis follows the resolution of the hazard and exposure datasets used as input. In
most cases these have the same resolution, and where this is not the case the tendency is either to resample the lower resolution
datasets to the higher resolution, or vice versa. Therefore, the resolution of the input hazard and exposure databases tends to
show the same overall patterns as those discussed above for the risk calculations. The most commonly used datasets used to
represent exposure are gridded datasets of population and GDP. Direct economic damage is further assessed using land use
data, whilst in recent studies related to earthquakes and tsunamis the use of capital stock estimates (based on building
typologies) has become more common. Given the importance of agricultural impacts for drought, exposure is also represented
using datasets such as gridded agricultural area, cropland area, planting area, and so forth. For wildfires, area of forest and
grassland are also used. Nevertheless, Kreibich et al. (2019) highlighted that especially for drought, data on losses/impacts that
are directly attributed to the hazard of drought are lacking.

Methods and datasets used to represent vulnerability are highly diverse. Within the flooding community, the most common
approach is to use intensity damage functions (IDFs). For flooding, the IDF takes the form of a depth-damage function. In
most studies, one global IDF is used (especially for coastal flooding), whilst for river flooding several studies have also used
regional or country level IDFs. Jongman et al. (2015) also use regional ratios of affected GDP to reported losses and affected
population to reported fatalities; the latter is also used by Dottori et al. (2018). Global IDFs are also used in studies that examine
wildfires. Some earthquake and tsunami studies have also used IDFs. They either use IDFs per income class or region (i.e. not
a global function), or in more recent years there has been a tendency to use IDFs related to building types. Another approach
is to use empirically derived regressions between reported impacts and a given levels of hazard/exposure to derive empirical
regression equations. In volcanology the limited amount of impact data has meant that only a few vulnerability and fragility
functions for physical vulnerability have been developed (Blong, 2003; Jenkins et al., 2014; Maqsood et al., 2014, Wilson et
al., 2014). Global TC and volcano studies tend to use various socioeconomic variables at country (or state) level as a proxy
of vulnerability. Therefore, the spatial representation of vulnerability is much coarser, and does not use gridded datasets in the
same way as hazard or exposure.





It is clear that the different hazard communities use some common datasets to represent exposure, for example relating to population and GDP. However, there are also differences and opportunities for learning. For example, the use of building stock data based on building typologies in earthquake and tsunami studies could provide opportunities for improving the assessment
of other hazards. The same can be said for the other hazards, where advances in IDFs related to building type offer opportunities outside the seismic community. On the other hand, approaches that have been developed to project future exposure in the flood risk community could be harnessed by the other hazard communities.

### 3.3. Risk indicators

The most commonly used risk indicator is the number of affected people, which is used in 59% of the reviewed studies. A
large number of studies also use some indicator of direct economic damage (44%), with fatalities (26%) and affected GDP (24%) have also been used in many studies. Fatalities have been much less commonly assessed in flood and drought-risk studies than in studies of the other hazards, offering potential for cross hazard learning on methods for fatality assessment.

### 3.4. Future DRR measures

To date, future DRR measures have only been explicitly included in several studies, all of which are related to flooding. Global
scale assessments in coastal flooding have been forward-looking from the outset, since the original studies were developed to assist climate adaptation studies. As such, they also include a wider range of DRR measures (both structural and nature-based) than the few river flood studies that have explicitly included DRR measures in recent years. The costs of DRR measures have only been explicitly assessed in a couple of studies, for coastal or river flood risk. Hence, in this regard the global flood modelling community has a lot of knowledge and examples that can be used to begin to include DRR measures in global scale
assessments of other hazards. To date, no global studies have assessed the influence that human behaviour and perception can have on the effectiveness of DRR measures.

### 3.5. Type of analysis

In Table 1, we have classed studies as either non-probabilistic (NP) or probabilistic (P), whereby probabilistic refers to studies that assess expected annual impacts either by integrating across return periods of based on a probabilistic stochastic event set.
For droughts, volcanoes, and landslides, studies to date have used non-probabilistic approaches only. Studies on floods and earthquakes have seen a move towards more probabilistic studies in more recent years, and the two studies reviewed for tsunamis also use a probabilistic approach. For wildfires and TCs, both approaches are used, with too few studies to be able to see any particular change in focus through time. A major difference between the studies of earthquakes and tsunamis, in comparison to the other hazards, is the extensive use of stochastic event sets in the former. Stochastic modelling could also be
beneficial for the assessment of other several other hazards, as discussed in Section 4.



## 4. Future research challenges and opportunities

Our review shows that the field of global natural hazard risk modelling has developed rapidly over the last decade, and advances continue to be made at a rapid pace. We show that there are differences between the modelling and assessment methods used in the different hazard communities, and show possibilities for learning between hazards. There are also opportunities for learning from methods and approaches being developed and applied to assess natural hazard risks at continental or regional scales. As section 2 demonstrates, rather than simply make direct comparisons, it is essential to contextualise the reasons for advances in global risk assessments for certain hazards compared with others, which include policy drivers (e.g. climate change), different levels of complexity (e.g. multi-hazard environments) and the relative frequency of occurrence of certain hazard related disasters. We have identified within the literature opportunities for addressing some of the key challenges. Given the constraints of space, this is not intended to be an exhaustive list, and is more intended to encourage dialogue on knowledge sharing between research and policy communities working on different hazards and at different spatial scales.

### 4.1. Hazard

Continual improvements in hazard modelling are required, both to correctly represent processes and to increase resolution. The availability of higher resolution input datasets with increased accuracy is helping in this regard and is a common theme across hazard studies. In some cases, global hazard models are available, but have not yet been used in global risk assessments. For example, Jenkins et al. (2015) provide a global volcanic hazard assessment (10 km x 10 km), that accounts for ash fall affected by local wind conditions. While the geographic extent of major disasters may be large, the specific hotspots for hazard may be much more localised. As such, there are ample opportunities to model the downscaled impact of hazards. For example, global models at present do not capture the downstream impact of landslide material (e.g. Nowicki Jesse et al. 2017), even though in many settings debris flows and sediment related damage can be the costliest type of hazard (Turowski et al. 2014). Within the global earthquake and tsunami risk modelling community, we see many examples of the use of stochastic events sets. Similar approaches could be developed for assessing risk of other hazards. For example, for the modelling of TCs, several models have been developed to generate synthetic TCs, such as the STORM model (Bloemendaal et al 2019), the MIT model (Emanuel and Nolan, 2004) or CHAZ (Lee et al 2018). Such TC events could then be used to force global storm surge models, thereby also benefitting global coastal flood risk assessment. Methods for generating large, synthetic event sets could also be especially useful for those events with high spatial and temporal resolution that currently miss global approaches, such as pluvial flooding and SCS-related perils.

There is also a tendency to focus on one parameter of the hazard, whilst a hazard's impact is often related to several parameters. For example, in flood risk analysis, global hazard studies focus on the flood depth, whilst risk is also related to other parameters such as flood duration, velocity, and the rate at which floodwaters rise (Ward et al., 2016). Similar issues exist for TCs, where there has been a lot of recent attention on their possible slowing down and stalling (Kossin, 2018; Wang et al., 2018; Hall and





Kossin, 2019). For example, Hurricane Dorian in 2019 stalled over the Bahamas for 36 hours, pounding large parts of the island with 270 km/h winds and 5 m storm surge.

For the water-related hazards, one avenue towards improved global hazard modelling is the improvement in hydrodynamic modelling of floods. For example, Sampson et al. (2015) present a fully hydrodynamic modelling approach for the globe, which could address some of the stated problems. The approach has been further developed by Wing et al. (2017) for the conterminous USA, and further been applied for current and future flood risk assessment in the USA at continental scale (Wing et al., 2018). For coastal flooding, the fully hydrodynamic model GTSM is now being used to simulate water levels due to

surge and tide up to the coastline, but then simple planar models are used to translate these water levels into inundation maps on land. Vafeidis et al. (2019) have shown the importance of accounting for hydrodynamic processes by developing an approach to assess the impacts of water-level attenuation due to different land covers on flood hazard. This approach can be used as a first step towards improving global coastal flood risk assessment. Vousdoukas et al. (2016) use the hydrodynamic LISFLOOD-FP model to assess coastal flood hazard at the European scale, and this has been applied for European scale coastal

flood hazard by Koks et al. (2019). Hydrodynamic inundation modelling is also being applied by Schäfer (2018) for the modelling of tsunami events. For the case of drought, studies typically focus only on a single type of drought/drought index. To comprehensively understand drought events and corresponding risks, an all angle view is of needed. Furthermore, for the global scale, remote sensing products that capture hazard and impact at the same time (NDVI, fAPAR) should be applied more.

## 4.2. Exposure

Similarly, continual developments are being made in the improvement of global exposure databases. As stated in the review, building typologies and/or investment data have been used to develop global databases of capital stock, which are now routinely used in global earthquake and tsunami modelling (Gunasekera et al., 2015). These data can also be applied to other hazard types in most cases, underlining the need for communication and collaboration across hazard communities. Efforts are also ongoing to develop exposure maps based on building material types within the flood risk community. For example,

Englhardt et al. (2019) have developed an approach for mapping exposure in urban and rural areas in Ethiopia, based on data on buildings and their materials. This method is currently being tested for several other countries in Africa. Pittore et al. (2017) discuss the challenges in designing a global and spatial-temporally dynamic exposure database, focusing on building stock. Other data sources, such as OpenStreetMap, also offer the opportunity to use building level information to improve global risk modelling. For the USA, Wing et al. (2018) have used the FEMA National Structure Inventory. The latter dataset is interesting

in that it is also accompanied by projected distributions under several future SSPs, whilst currently, forward-looking projections of exposure at global level are limited to GDP, population, and land use. Recent studies have also shown the importance of examining temporal variations in exposure (e.g. between day and night and between seasons) (e.g. Freire et al., 2015) and between different income groups (e.g. Winsemius et al., 2018; Hallegatte et al., 2016). The importance of capital stock models such as those which encompass buildings, infrastructure, cross-sector applications and contents are shown in the



GRADE process; given that in most cases the building stock is only one portion of the capital stock at risk (Gunasekera et al., 2018).

### 4.3. Vulnerability

As evidenced from the review, much attention is required to improve the representation of vulnerability in global risk models. Currently there exist a limited number of vulnerability and fragility functions for some hazards (see Murnane et al. 2019),

compounded by the limited amount of impact data available to inform them (e.g. volcanic eruptions). However, a limited number of socioeconomic vulnerabilities in volcanic environments are being considered in – for instance – global datasets (e.g. global fatalities, Brown et al., 2017) and there are some studies accounting for the indirect impacts of eruptions (loss of livelihood, displacement and resettlement) at the volcano scale (e.g. Barclay et al., 2019). In time and with more resources and studies, such efforts may be scalable, and used to inform future global risk studies.

The highly temporal and spatial dynamics of vulnerability and the resulting non-linearity of risk has been underscored by UNDRR's Global Platform for Disaster Risk Reduction. While recent studies at regional and local scale have begun to account for these aspects in changing hazard (e.g. Mora et al. 2018) and exposure conditions (e.g. Cammerer et al. 2013), only a few studies account for the dynamics of vulnerability (e.g. Kreibich et al., 2015) across multiple hazards. Cutter and Finch (2008) have assessed future spatial and temporal patterns of social vulnerability at a national scale based on historical events. For

future projections, climate change is widely recognized as an important driver of the increased frequency and intensity of weather-related hazards, but does not explain the (projected-) changes in damages caused by geophysical hazards such as earthquakes. An improved understanding of future vulnerability can significantly improve the ability of risk managers to more efficiently implement DRR measures. Recent studies at the continental or global scale show that vulnerability to natural hazard-related disasters is decreasing in some areas, because people adapt over time and reduce vulnerability (e.g. Ciscar et al. 2019;

Jongman et al. 2015). In other areas, future vulnerability is expected to increase, for example due to a limited availability of resources to adapt (Winsemius et al., 2018) or due to the impacts of successive disasters that push communities into poverty (Mirza 2003).

Some of the remaining scientific challenges include a harmonisation of indicators used to assess damages across a wide range of different hazard types, in order to enable the collection of loss data that is comparable across hazards. This would allow for

a better comparison of the dynamics of vulnerability between different hazards. Currently, the impact data that are collected by countries, first responders, and researchers from different fields remain very heterogeneous (Cutter et al. 2015; AghaKouchak et al. 2018) and the data are often collected at different times after a disaster.

### 4.4. DRR measures

The number of global risk studies that explicitly include DRR measures is extremely limited, and limited to flood risk studies,

especially coastal flooding. Even then, most of these studies have assessed structural measures and do not explicitly examine the costs. Much can be learnt from studies at local to regional scale, and it is certainly beyond the scope of this paper to provide a review of the many studies addressing DRR at this scale. A specific aspect that has not been covered in any of the global risk



studies reviewed is the influence that human behaviour and perception can have on the effectiveness of DRR measures, through various feedbacks. A classic example in hydrology is the levee effect (White, 1945), in which increased levels of flood

protection from levees and dikes can also lead to increased exposure and/or vulnerability in areas protected by dikes. Similarly, for wildfires feedbacks exist between the physical risk and human actions to attempt to manage fuel and suppress fires. A promising way to address these feedbacks is through agent-based models that attempt to represent the behaviour of agents (e.g. individuals, businesses, governments) through a set of decision rules (e.g. Aerts et al., 2018). An application of the ABM approach has recently been used in natural hazard risk modelling at the continental scale (Haer et al., 2019), paving the way

towards exploring the use of these methods at global scale. Another aspect that is often overlooked, especially on a global scale, is the interactions between different DRR measures that are aimed at specific hazards (Zaghi et al. 2016; Scolobig et al., 2017). Moreover, assessing the impact of complex multi-hazard damages that result from hazard chains (e.g. an earthquake followed by a flood) has only been assessed at local scales, for example by using a probabilistic approach to calculate the probabilities of different final damage states (e.g. Korswagen et al. 2019). These complex hazard chains require the design of

structures and DRR measures that are able to address the combined damages of different hazard chains (Korswagen et al. 2019).

Studies that consider the dynamics of how drivers impact risk into the future also enable the assessment of prospective DRR actions across exposure and vulnerability components. By modelling exposure profiles dynamically, risk reduction actions that consider where development occurs and how this can be changed to reduce future losses can be assessed. This would support

demonstrating the effectiveness of land use planning and risk-sensitive developments as a DRR action. Similarly, for vulnerability, incorporating assessment of future vulnerability and the inclusion of improved building standards allows for demonstrating the benefits of more resilient construction. If changes in exposure and vulnerability are excluded from disaster risk modelling, then the assessment of prospective DRR measures is extremely challenging. Examples of this globally however are limited. Regional and city-level studies showing elements of these benefits have been demonstrated for multiple risks using

coupled hazard models with cellular-automata land use models and capital stock models (Lallemant, 2015; Riddell et al., 2019;). Global dynamic models of urban and land use change however do exist and efforts could be made to effectively couple these with global hazard and risk models (Hasegawa et al., 2017; Van Asselen and Verburg, 2013).

### 4.5. Multi-hazard and multi-risk

There is a rapidly growing policy and scientific recognition and dialogue on the need for multi-hazard (both the multiple

hazards and the simultaneous, cascading or cumulative occurrence of these; UNDRR 2017) risk assessments, as exemplified by high-level discussions at the UNDRR Global Platform 2019 and the aims of the Sendai Framework. These call upon the science community for an increased understanding of the risk of consecutive and cascading disasters (UNDRR, 2019). So far, the vast majority of global risk assessment studies have examined risk from a single hazard type, and indeed have examined a single parameter of the hazard (section 4.1). However, many environments are multi-hazardous and many hazards can trigger

secondary or cascading hazards. For example, volcanoes can produce a wide variety of primary and secondary hazards that



can occur simultaneously or sequentially, and that differ widely in their spatial extent, duration, dynamic characteristics and associated impacts. Capturing all these hazards, and their impacts, within the one assessment is very challenging, and typically the future hazard or risk is considered separately for each type of hazard (e.g. Sandri et al., 2014) or by assuming a given eruption scenario (e.g. Lindsay and Robertson, 2018). Methods for volcanic multi-hazard assessment across multiple scenarios,

where the range of potential future volcanic hazards are shown on the one map, have been developed but as yet only been applied at the single volcano scale (e.g. Neri et al., 2013).

Interactions between different primary hazards can also influence the overall risk (Gill and Malamud, 2014; Korswagen et al., 2019). For example, within the flooding community, there has recently been much attention for so-called compound floods, whereby the interaction of coastal, river, and pluvial floods can influence the overall hazard and risk (Zscheischler et al., 2018).

Methods are being developed at the global scale to assess both the statistical dependence between these hazards (e.g. Ward et al., 2018; Bevacqua et al., 2019; Couasnon et al., 2019) and their physical impacts in terms of hazard (Ikeuchi et al., 2017), with the step towards risk being the next logical step.

Understanding inter-hazard linkages is also important for properly calibrating estimates of risk in the aftermath of major disasters. Earthquakes in particular can affect the long-term propensity of a given landscape to fail via landsliding (Marc et al.

2015), while the prior saturation state of a landscape - due to flooding or human input of irrigation water - can increase landslide susceptibility during an earthquake itself (Bradley et al. 2019; Watkinson and Hall 2019). These interactions remain poorly constrained, but can influence the long term recovery from major hazards.

Several studies have identified the current shortcomings in future exposure and vulnerability projections for multi-hazard risk assessments (e.g. Gallina et al., 2016). In spite of the clear need to adopt a multi-hazard risk approach to global risk

assessments, the very nature of the endeavour (accounting for multiple, interrelated hazards and their dynamic influence on vulnerability and exposure) has arguably restricted progress towards truly comprehensive analyses of global multi-hazard risk. There is a need to address the previously mentioned challenges with critical work at multiple scales (local to global) towards comprehensive global multi-hazard risk assessments. Collaboration between the international hazard science and risk research communities is key to progress.

### 4.6.   Use of citizen science/crowdsourced data

All elements of risk may be better quantified or qualified using citizen science and crowd-sourced data, which are utilised across a number of natural hazards (Hicks et al., 2019). De Bruijn et al. (2017) already showed that passively shared information through social media platforms can provide a plethora of qualitative and sometimes quantitative information of flood hazards, as well as impacts in near real-time. Also, active sharing mechanisms are becoming available. For example,

citizen scientists report information on landslide hazard that is otherwise difficult to obtain. At the global scale, volunteers can contribute information through Landslide Reporter (https://landslides.nasa.gov/reporter). Numerous local, regional, or national crowdsourcing projects have also been undertaken (Juang et al. 2019). Information on landslide timing is often missing from existing landslide inventories, because neither remote sensing nor geologic fieldwork are determining this feature (Kocaman





and Gokceoglu, 2018). However, precise knowledge of landslide timing is crucial for research into the triggering mechanisms

of landslides. Citizen scientists could remedy this in cases where they have first-hand knowledge of recent events. Citizen science has been applied in volcanic environments, from observations of ash fall (Wallace et al., 2015) to community-based monitoring (Stone et al., 2014), and has a well-established application in earthquakes (e.g. USGS "did you feel it").

An exciting avenue in citizen science lies in more active mechanisms to report on hazards and their impacts (damages, households affected, people in need of help, and so on). In particular, where such reports concerns citizen's own property or

surroundings, they are likely to be more motivated and more accurate in their reporting and by having an active reporting mechanism, this can lead to a more structured approach to monitor and keep track of past hazards and their impacts. Opportunities lie particularly in reporting of local flash floods, local landslides and local drought conditions such as small reservoir states for rainfed farming, and grass states for pastoralists. These natural hazard conditions concern time and space scales that are too small to capture with other means, but may occur frequently and in many locations. Structured and organised

data collection by citizens is taking place more and more. Large-scale community mapping projects are initiated that rapidly increase the availability of well-organised taxonomic information on buildings and infrastructure (e.g. Soden and Palen, 2014; Iliffe et al., 2017). These may serve as exposure and vulnerability databases for multiple natural hazards, as well as drainage information, that can be used to establish flood hazard models (Winsemius et al., 2019). The tools and data platforms to collect, store and share such data in any resource setting are broadly available, such as OpenDataKit (Brunette et al., 2013) and

OpenStreetMap (Haklay and Weber, 2008). These data may be used for instance to train machine learning algorithms that estimate exposure and vulnerability characteristics based on remote sensing, or to keep risk models in rapidly changing environments, such as growing urban centres, up to date (Winsemius et al., 2019). These opportunities have been explored only to a limited extent and we foresee a growth in demand of these research directions.

### 4.7. Concluding remarks

As shown by this review, efforts to assess and map natural hazard risk at the global scale have increased considerably in the last decade, in an attempt to contribute to the Sendai Framework's first Priority for Action of Understanding Disaster Risk. This paper presents a first attempt to review these studies across different hazards, thereby examining similarities and differences between the approaches taken across the different hazards, and identifying potential ways in which different hazard communities can learn from each other. We also indicate several opportunities for addressing some of the pressing challenges

in global risk modelling. We hope that this review paper can serve to further encourage the dialogue on knowledge sharing between scientists and communities working on different hazards and at different spatial scales that has been facilitated by the session *Global and continental scale risk assessment for natural hazards: methods and practice* at the EGU General Assembly since 2012.



**Author contribution**

All authors contributed to reviewing the literature and writing the paper. PJW furthermore coordinated the review and writing process.

**Competing interests**

The authors declare that they have no conflict of interest.

**Acknowledgements**

PJW and NB received funding from the Dutch Research Council (NWO) in the form of a VIDI grant (grant no. 016.161.324) and VICI grant (grant no. 453.13.006). VB is funded by the DRIe.R- project, financed by the Ministry of Science, Research and the Arts of the State of Baden-Württemberg. RE is supported by an appointment to the NASA Postdoctoral Program at Goddard Space Flight Center.

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

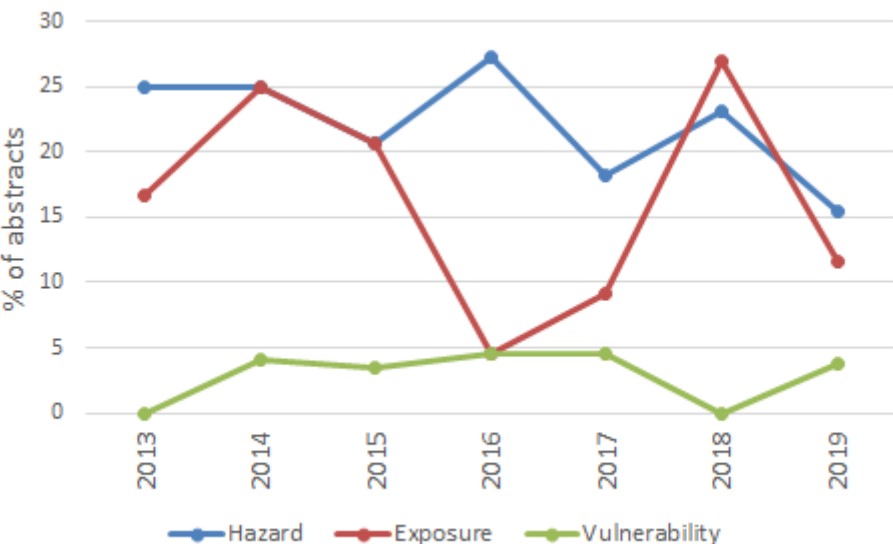

**Figure 1: Percentage of abstracts accepted and presented at the EGU session 'Global and continental scale risk assessment for natural hazards: methods and practice' explicitly mention examining future projections of risk based on future scenarios of hazard, exposure, or vulnerability projections.**


**Table 1: Summary of findings across the reviewed literature and across the reviewed aspects. Explanatory notes: 'Risk elements' - white of this risk element is not included, orange if included and static, green if included and dynamic; 'Type of risk analysis' - NP = probabilistic, P = probabilistic (P), S = stochastic event sets, RP = return period maps, Y = maps of yearly hazard, M = maps of monthly hazard, PE = past events, V = radius around specific volcano.**

330

| | Risk elements (Hazard / Exposure / Vulnerability) | Resolution of risk elements | | | Risk indicators (Direct damage / Indirect damage / Affected people / Affected GDP / Fatalities / Other) | | | | | | Future DRR measures (Structural / Nature-based / Non-structural / Costs / Behaviour) | | | | | Risk analysis | | |
|---|---|---|---|---|---|---|---|---|---|---|---|---|---|---|---|---|---|---|
| | | Hazard | Exposure | Vulnerability | Direct damage | Indirect damage | Affected people | Affected GDP | Fatalities | Other | Structural | Nature-based | Non-structural | Costs | Behaviour | Type | Time-horizon | Resolution |
| **River floods** | | | | | | | | | | | | | | | | | | |
| Kleinen & Petschel-Held (2007) | | Large basins | Large basins | | | | X | | | | | | | | | NP-R | 2080 | Large basins |
| Hirabayashi & Kanae (2009) | | 1.0° | 1.0° | | | | X | | | | | | | | | NP-Y | Yearly to 2100 | 1.0° |
| Jongman et al. (2012) | | 30" | 30" | | X | | X | | | Urban area exposed, assets exposed | | | | | | NP-R | 2050 | 30" |
| Hirabayashi et al. (2013) | | 2.5' | 2.5' | | | | X | | | | | | | | | NP-Y | Yearly to 2100 | 2.5' |
| Ward et al. (2013) | | 30" | 30" | Global IDF | X | | X | X | | Affected agricultural value | | | | | | P-R | Current | 30" |
| Arnell & Lloyd-Hughes (2014) | | 0.5° | 0.5° | | X | | X | | | | | | | | | NP-R | 2050, 2080 | 0.5° |
| UNISDR (2015a) | | 1 km | 5 km | Regional IDFs | X | | | | | | | | | | | P-R | Current | 1 km |
| Jongman et al. (2015) | | 30" | 30" | Regional fatality & loss ratios | X | | | | X | | | | | | | P-R | 2030, 2080 | 30" |
| Arnell & Gosling (2016) | | 0.5° | 0.5° | | X | | X | | | Affected cropland area | | | | | | NP-R | 2050 | 0.5° |
| Winsemius et al. (2016) | | 30" | 30" | Global IDF | X | | | | | | X | | | | | P-R | 2030, 2080 | 30" |
| Ward et al. (2017) | | 30" | 30" | Global IDF | X | | | | | | X | | X | | | P-R | 2080 | 30" |
| Alfieri et al. (2017) | | 30" | 30" | Country IDF | X | | X | | | | | | | | | P-R | 1.5, 2 & 4°C warming | 30" |
| Dottori et al. (2018) | | 2.5' | 2.5' & 7.5' | Country IDF & fatality ratios | X | X | X | | X | | | | | | | P-R | 1.5, 2 & 3°C warming | 2.5' |




| Study | | | | | | | | | | | Proj | Year | Resolution |
|---|---|---|---|---|---|---|---|---|---|---|---|---|---|
| Willner et al. (2018) | | 2.5' | 2.5' | | | X | | | X | | P-R | 2040 | 0.25° |
| **Coastal floods** | | | | | | | | | | | | | |
| Hoozemans et al. (1993) | | Country | Country | | | X | | Wetland loss & rice production | X | | NP-R | 1 meter SLR | Country |
| Hinkel & Klein (2009) | | Coastal segments | Coastal segments | Global IDF | X | X | X | Wetland loss | X | X | P-R | 2100 | Coastal segments |
| Hinkel et al. (2010) | | Coastal segments | Coastal segments | Global IDF | X | X | X | Wetland loss | X | X | P-R | Up to 5°C warming | Coastal segments |
| Jongman et al. (2012) | | 30" | 30" | | X | X | | Urban area exposed, assets exposed | | | NP-R | 2050 | 30" |
| Hallegatte et al. (2013) | | Coastal segments | Coastal segments | Global IDF | X | | | | X | X | P-R | 2050 | 90 m |
| Hinkel et al. (2014) | | Coastal segments | Coastal segments | Global IDF | X | X | X | Wetland loss | X | X | P-R | 2100 | Coastal segments |
| Fang et al (2014) | | 1 km | 1 km | | | X | X | | | | NP-S | Current | 1 km |
| Muis et al. (2016) | | 30" | 30" | | | X | | | | | NP-R | Current | 30" |
| Muis et al. (2017) | | 30" | 30" | | | X | | | | | NP-R | Current | 30" |
| Schuerch et al. (2018) | | Coastal segments | Coastal segments | | | | | Wetland loss | X | X | P-R | 2100 | Coastal segments |
| Hunter et al. (2019) | | City | City | Global IDF | X | | | | | | P-R | Current | City |
| Beck et al. (2019) | | 30" | 30" | Global IDF | X | X | X | | | X | P-R | 2100 | 90 m |
| Vafeidis et al. (2019) | | Coastal segments | Coastal segments | Global IDF | X | X | X | Area exposed | | | P-R | 2100 | Coastal segments |
| **Tropical cyclones** | | | | | | | | | | | | | |
| Peduzzi et al. (2009) | | 5 km | 5 km | Country socioeconomic variables | | X | | | | | NP-R | Current | 5 km |



| Reference | | | County (USA) & country (rest of world) | Income per county (USA) & country (rest of world) | | Impact | | Type | Period | Storm tracks |
|---|---|---|---|---|---|---|---|---|---|---|
| Mendelsohn et al. (2012) | | Storm tracks | | | X | | | P-S | 2100 | Storm tracks |
| Peduzzi et al. (2012) | | 2km | 30" | Country socioeconomic variables | X X X | | | NP-Y | 2030 | 30" |
| Cardona et al. (2014) | | 1km | 5 km | Regional IDFs | X | | | P-R | Current | 1 km |
| Fang et al. (2015) | | 30" | 30" & 0.5° | | X X | | | NP-R | Current | 0.1° |
| **Droughts** | | | | | | | | | | |
| Dilley et al. (2005) | | 2.5° | 2.5' | | X X X | Road infrastructure | | NP-Y | Current | 2.5' |
| Li et al. (2009) | | 0.5° | Country | Country socioeconomic variables | | (Reduced) yield of maize, wheat, rice, barley | | NP-Y | 2050, 2100 | 0.5° |
| Arnell et al. (2013) | | 0.5° | 0.5° | | | Cropland exposed, wheat and soybean productivity | | NP-M | 2030, 2050, 2080, 2100 | 0.5° |
| Christenson et al. (2014) | | 2.5° | 5' & 0.5° | 0.5° regressions | X | | | NP-Y | Current | 2.5' |
| Yin et al. (2014) | | 0.5° | 0.5° | | | Yield of maize | | NP-RP | Current | 0.5° |
| Carrão et al. (2016) | | 0.5° | 30" & 5' | 5' to country to socioeconomic variables | | "Index" across several factors | | NP-Y | Current | 0.5° |
| Guo et al. (2016) | | 0.5° | 30" | 0.5° regressions | | (Reduced) yield of maize | | NP-RP | Current | 0.5° |
| Smirnov et al. (2016) | | 2° | 2° | | X | | | NP-M | 20 year time-slices to 2100 | 2° |
| Arnell et al. (2018) | | 0.5° | 0.5° | | X | Cropland exposed | | NP-M/Y | 1.5 & 2°C warming | 0.5° |
| Liu et al. (2018) | | 0.5° | 0.5° | | X | | | NP-M | 1.5 & 2°C warming | 0.5° |
| **Wildfires** | | | | | | | | | | |
| Cao et al. (2015) | | 1 km | 1 km | Global functions | | Net primary production | | NP-Y | Current | 1 km |




| Reference | | | | Vulnerability | | | | | | | Type | Time | Resolution |
|---|---|---|---|---|---|---|---|---|---|---|---|---|---|
| Meng et al. (2015) | | 0.1° | 0.1° | Global functions | | | | | Forest area burnt | | NP-R | Current | 0.1° |
| Shi et al. (2015) | | 0.1° | 0.1° | Global functions | | | | | Forest area burnt | | P-R | Current | 0.1° |
| Knorr et al. (2016) | | 1° | 0.5° | | | X | | | Forest area burnt | | NP-Y | 1901 - 2100 | 1° |
| **Earthquakes** | | | | | | | | | | | | | |
| Chan et al. (1998) | | 0.5° | 0.5° | Income class functions | X | | | | | | NP-S | Current | 0.5° |
| Dilley et al. (2005) | | 2.5' | 2.5' | Global fatality ratios | | X | X | X | | | NP-R | Current | 2.5' |
| Jaiswal & Wald (2010) | | 1 km | 1 km | Regional fatality ratios | | X | X | | | | NP-S | Current | 1 km |
| Jaiswal & Wald (2011) | | 2 km | 1 km | Regional loss functions | X | | X | | | | NP-S | Current | 1 km |
| UNISDR (2013, 2015, 2017) | | 5 km | 5 km | Building type IDFs | X | | | | | | P-S | Current | 1 km |
| Daniell (2014); Daniell & Wenzel (2014) | | 1 km | 1 km | Country regressions | X | X | X | X | X | | NP-S | Current | 1 km |
| Li et al. (2015) | | 0.1° | 0.1° & 0.5° | Regional fatality rates and damage functions | X | | X | X | X | | P-R | Current | 0.1° & 0.5° |
| Silva et al. (2018) | | 1km | variable | Building type IDFs | X | | | | | | P-S | Current | Variable |
| **Tsunamis** | | | | | | | | | | | | | |
| Løvholt et al. (2015) | | 90 m | 1 km | Building type IDFs | X | X | | X | | | P-S | current | 1 km |
| Schäfer (2018) | | 90 m | 90 m | Building type IDFs and global fatality function | x | | x | x | x | | P-R | current | 90 m |
| **Volcanoes** | | | | | | | | | | | | | |
| Yokoyama et al. (1984) | | Volcano | Volcano | Indices per volcano | | | | | Risk index | | NP-PE | Current | Volcano |
| Small & Naumann (2001) | | | 2.5' | | | x | | | | | NP-V | Current | Volcano |
| Dilley et al. (2005) | | 2.5° | 2.5' | Regional fatality & loss ratios | | X | X | X | | | NP-PE | Current | 2.5' |


| Reference | | | | | | | | | | | |
|---|---|---|---|---|---|---|---|---|---|---|---|
| GAR 2015 (Auker et al., 2015; Brown et al., 2015a,b) | | Volcano | Volcano | | | x | x | | NP-PE | Current | Volcano |
| Pan et al. (2015) | | Volcano | 30" | Global fatality function | | | X | | NP-PE | Current | Volcano |
| Freire et al. (2019) | | | 250 m | | | x | | | NP-V | Current | Volcano |
| **Landslides** | | | | | | | | | | | |
| Dilley et al. (2005) | | 30" | 2.5' | Country regressions | X | | X | X | | NP-Y | Current | 2.5' |
| Nadim et al. (2006) | | 30" | 30" | Country regressions | | | x | | NP-Y | Current | 30" |
| Yang et al. (2006) | | 0.25° | 0.25° | Country regressions | | | X | | NP-Y | Current | 0.25° |
| Nowicki Jessee et al. (2017) | | Variable | 30" | | | x | | | NP-Y | Current | Variable |