# Peer review of "Review article: Natural hazard risk assessments at the global scale"

_Natural Hazards and Earth System Sciences, 2019_

## Short Comment (SC1) · 28 Dec 2019

(1) In terms of the exposure of earthquake (line 426), some important progresses are neglelected and strongly encouraged to be added.

Djordjević, M., Radivojević, A., Dragović, R. and Filipović, I., 2016. EXPOSURE TO EARTHQUAKES-DISTRIBUTION AND CHANGE OF THE WORLD'S POPULATION WITH REGARD TO DISPOSITION OF SEISMIC ACTIVITIES. Journal of the Geographical Institute'Jovan Cvijic'SASA, 66(3).

Pesaresi, M., Ehrlich, D., Kemper, T., Siragusa, A., Florczyk, A.J., Freire, S. and Corbane, C., 801 2017. Atlas of the Human Planet 2017.

(2) For the future studies on the changes (dynamics) of exposure at the global or re-

gional scale, the deficiency on this topic for geological hazard will also be an interesting opportunity (line 705, section 4.2). For example, the changes in population exposure to earthquake hazard have revealed that urbanization and related migration played an important roles in increasing the number of vulnerable people to earthquake hazard in Asia (Dou et al., 2018 ) and in China (He et al., 2016; Huang et al., 2019). I believe that these progress would be important in the context of global urbanization and SDG11 (sustainable cities and communities).

Yinyin Dou, Qingxu Huang, CHunyang He, Shiting Meng, Qiang Zhang, 2018, Rapid Population Growth throughout Asia's Earthquake-Prone Areas: A Multiscale Analysis, International Journal of Environmental Research and Public Health, 15(9): 1893

Chunyang He, Qingxu Huang, Yinyin Dou, Wei Tu, Jifu Liu, 2016, The population in China's earthquake-prone areas has increased by over 32 million along with rapid urbanzation, Environmental Research Letters, 11: 074028

Qingxu Huang, Shiting Meng, Chunyang He, Yinyin Dou, Qiang Zhang, 2019, Rapid Urban Land Expansion in Earthquake-Prone Areas of China, International Journal of Disaster Risk Science, 10(1): 43-56

---

## Referee Comment (RC1) · Francesco Dottori (Referee) · 5 Feb 2020

This review paper provides a valuable and comprehensive overview of the state of the art of global risk models for natural hazards. I much agree on the objective of comparing modelling approaches across sectors, and I believe such comparison may offer a contribution towards the improvement of global models. The paper is generally well structured. The sections addressing the different hazards are balanced and informative, including the supplementary material.

Before recommending the paper for publication, I'd like to suggest some minor changes:

Section 2.1.1, River floods: several modelling frameworks only cover large river basins

(e.g. 5000km2 in Alfieri et al., 2017) while minor river network is not considered (even though there are exceptions). I think it would be important to mention this as a general limitation.

Section 2.1.3, Pluvial floods: it would be good to shortly discuss the issue of modelling flash floods at global scale (i.e. fast-developing flood events occurring in the minor river network). Maybe it's worth mentioning here the global flood model by Sampson et al. (2015), because it is the only global flood model including a pluvial flooding component (to my best knowledge).

Section 2.2 I would mention the multi-hazard nature of tropical cyclones (i.e. the fact that impacts are caused by strong winds and the combination of pluvial, coastal, river flooding).

Line 297 typo: "Commonly, drought hazard is defined as..."

Line 301 typo: "Hence, a universal definition of drought seems impracticable..."

Line 693-694: It's worth mentioning that the study by Wing et al. (2018) also evaluated risk from pluvial flooding.

Section 4.3, Vulnerability: another important challenge here is the reliability of existing global loss datasets, which have known limitations in data coverage, accessibility, completeness and accuracy (e.g. see UNISDR-CRED (2018) related to EM-DAT database). These limitations hamper the validation of any large-scale modelling framework and I think they should be mentioned, either in section 4.3 or in a dedicated section.

Table 1: please consider the idea of separating each hazard in a dedicated table. Also, please define the meaning of IDF in the caption.

Additional References

UNISDR-CRED (2018). Economic Losses, Poverty and Disasters 1998-2017. available at https://www.unisdr.org/2016/iddr/IDDR2018_Economic%20Losses.pdf

---

## Referee Comment (RC2) · 10 Feb 2020

Thank you for the opportunity to review this valuable & timely effort. For myself & my colleagues at the World Bank, this catalogue will be useful to support and employ risk analytics at global scale. For each of the hazards, the overview of the current state of disaster risk modeling is detailed and comprehensive, and the article's structure will make it a handy reference.

In the discussion of future research challenges & opportunities in section 4, I suggest that the authors include as a subsection a brief mention of work on socio-economic vulnerability and resilience to disasters, at global level. Although socio-economic characteristics are typically left out of the canonical hazard/exposure/vulnerability heuristic

(line 44), state-of-the-art risk analytics are increasingly moving to include them, as described by the authors throughout the text.

For example, socio-economic heterogeneities are noted as relevant inputs to drought (lines 316, 344) and wildfire risk (lines 346, 356-8), and would likely also be useful to develop representations of landslides (lines 520-24). Deep interdependencies between these hazards' impacts and socio-economic risk factors may be one reason that modeling of these hazards is still relatively rudimentary, suggesting an opportunity for further research.

Similarly, for hazards that have proved more tractable including flooding and earthquakes: the efficacy of DRR measures can depend greatly on socio-economic feedbacks including risk perception (lines 645, 753), social vulnerability (734), and resilience (740). Incorporating these datasets and dynamics will be essential to the development of CBA tools.

As risk analytics achieve greater spatial resolution and theoretical sophistication, socio-economic information layers are increasingly relevant for risk analytics, and essential for DRR applications (even apart from political economy considerations). For these reasons, and to summarize as an opportunity for research the point made throughout the text, I recommend that the authors include a brief note in Section 4. I know that the authors are familiar with this area from their own research, and I suggest as well my colleagues' work on the subject at global scale.

2 additional notes: (line 297) "is" misspelled as "us" (lines 792 & 795) replace "between" with "among"

Ward PJ, Jongman B, Aerts JCJH, Bates PD, Botzen WJW, Diaz Loaiza A, Hallegatte S, Kind JM, Kwadijk J, Scussolini P and Winsemius HC (2017) A global framework for future costs and benefits of river-flood protection in urban areas. Nature Climate Change 7, 642–646.

Hallegatte, Stephane, et al. Shock waves: managing the impacts of climate change on poverty. The World Bank, 2015.

Hallegatte S, Bangalore M and Vogt-Schilb A (2016) Assessing socioeconomic resilience to floods in 90 countries. World Bank Policy Research Working Paper No. 7663. The World Bank, Washington, DC. Available at https://ssrn.com/abstract=2781020.

Hallegatte S, Vogt-Schilb A, Bangalore M and Rozenberg J (2017) Unbreakable: Building the Resilience of the Poor in the Face of Natural Disasters. Washington, DC: World Bank.
* * *

---

## Author Comment (AC1) · 20 Feb 2020

We would like to thank you for taking the time to provide your valuable short comment on our paper. In the text below, we respond to each comment one by one.

(1) In terms of the exposure of earthquake (line 426), some important progresses are neglected and strongly encouraged to be added (Djordjevic et al., 2016; Pesaresi et al., 2017). We thank for the reviewer for the comment and suggested literature. We will clarify in the revised manuscript that the references included in the table are those that do not use the index-based methods or overlays of a single hazard map with exposure data (e.g. population, GDP) to assess global exposure to earthquake hazard. However, propose to add these references to the Supplementary Information in the following sentence: "Several global assessments used index-based methods or overlays of a single hazard map with exposure data (e.g. population, GDP) to assess global exposure to earthquake hazard (e.g. Davidson and Shah, 1997; Cardona, 2005; Hopkins, 2009; Peduzzi et al., 2009; Cardona and Carreño, 2011, Djordjević et al., 2016; Pesaresi et al., 2017)".

(2) For the future studies on the changes (dynamics) of exposure at the global or regional scale, the deficiency on this topic for geological hazard will also be an interesting opportunity (line 705, section 4.2). For example, the changes in population exposure to earthquake hazard have revealed that urbanization and related migration played an important roles in increasing the number of vulnerable people to earthquake hazard in Asia (Dou et al., 2018 ) and in China (He et al., 2016; Huang et al., 2019). I believe that these progress would be important in the context of global urbanization and SDG11 (sustainable cities and communities). We thank the reviewer for highlighting these very interesting papers. There are indeed studies at local, regional, and continental scales that show this important signal across different hazard types. Unfortunately, we are unable to include all of these references in the manuscript due the focus on global scale models – expanding to all regional scale assessments would make the paper prohibitively long.

References cited by the reviewer or in our response to the reviewer • Djordjević, M., Radivojević, A., Dragović, R., Filipović, I., 2016. Exposure to earthquakes – distribution and change of the world's population with regard to disposition of seismic activities. J. Geogr. Inst. Cvijic., 66, 353-370, doi: 10.2298/IJGI1603353D • Pesaresi, M., Ehrlich, D., Kemper, T., Siragusa, A., Florczyk, A.J., Freire, S. and Corbane, C., 801 2017. Atlas of the Human Planet 2017. • Dou, Y., Qingxu Huang, CHunyang He, Shiting Meng, Qiang Zhang, 2018, Rapid Population Growth throughout Asia's Earthquake-Prone Areas: A Multiscale Analysis, International Journal of Environmental Research and Public Health, 15(9): 1893 • He, C., Qingxu Huang, Yinyin Dou, Wei Tu, Jifu Liu, 2016, The population in China's earthquake-prone areas

has increased by over 32 million along with rapid urbanzation, Environmental Research Letters, 11: 074028 • Huang, W., Shiting Meng, Chunyang He, Yinyin Dou, Qiang Zhang, 2019, Rapid Urban Land Expansion in Earthquake-Prone Areas of China, International Journal of Disaster Risk Science, 10(1): 43-56

―――――――――――――――――

---

## Author Comment (AC2) · 20 Feb 2020

This review paper provides a valuable and comprehensive overview of the state of the art of global risk models for natural hazards. I much agree on the objective of comparing modelling approaches across sectors, and I believe such comparison may offer a contribution towards the improvement of global models. The paper is generally well structured. The sections addressing the different hazards are balanced and informative, including the supplementary material. Before recommending the paper for publication, I'd like to suggest some minor changes. We thank the reviewer for the time taken to review our manuscript and for his useful comments. We are pleased that the reviewer finds the manuscript to be a useful contribution, well-balanced, and informative. In preparing a revised manuscript, we will take care to address the minor

comments of the reviewer. We respond each of the individual review comments below.

(1) Section 2.1.1, River floods: several modelling frameworks only cover large river basins (e.g. 5000km2 in Alfieri et al., 2017) while minor river network is not considered (even though there are exceptions). I think it would be important to mention this as a general limitation. Indeed, this is an important point to add. We propose to add the following statement to section 2.1.1. "It should be noted that each of the models described here has its own minimum catchment size (ranging from ~500 to ~5000km2), under which hazard (and therefore risk) are not calculated"

(2) Section 2.1.3, Pluvial floods: it would be good to shortly discuss the issue of modelling flash floods at global scale (i.e. fast-developing flood events occurring in the minor river network). Maybe it's worth mentioning here the global flood model by Sampson et al. (2015), because it is the only global flood model including a pluvial flooding component (to my best knowledge). Thanks for the very valuable suggestion. We propose to add the following text to section 2.1.3: "Sampson et al. (2015) do include floods in small river channels (with catchment less than 50km2) driven by intense local precipitation. To do this, they use a 'rain-on-grid' method in which flow is generated by simulating rainfall directly on the DEM at a high resolution (3" x 3"), using Intensity-Duration-Frequency relationships of extreme rainfall from ~200 locations around the world. However, they state that it is not known whether this method provides robust estimates of return period rainfall globally, and also indicate the importance of tackling the aforementioned difficulties. Wing et al. (2018) use this method to assess flood hazard and risk in the conterminous USA."

(3) Section 2.2 I would mention the multi-hazard nature of tropical cyclones (i.e. the fact that impacts are caused by strong winds and the combination of pluvial, coastal, river flooding). Thank you for the suggestion. We propose to emphasise this with the following sentence in section 2.2: "A defining aspect of a TC hazard is that it is composed of wind, precipitation, and storm surge, and the impacts result from a combination of these. However, the current studies to date do not explicitly model all of these aspects.".

This is also further elaborated on in section 4.1 as part of the discussion.

(4) Line 297 typo: "Commonly, drought hazard is defined as..." Thank you. We have amended the sentence as suggested.

(5) Line 301 typo: "Hence, a universal definition of drought seems impracticable..." Thank you. We have amended the sentence as suggested

(6) Line 693-694: It's worth mentioning that the study by Wing et al. (2018) also evaluated risk from pluvial flooding. We propose to add the following statement to section 2.1.3: "Wing et al. (2018) use this method to assess current flood hazard and risk in the conterminous USA." (see also response to reviewer's comment (2)).

(7) Section 4.3, Vulnerability: another important challenge here is the reliability of existing global loss datasets, which have known limitations in data coverage, accessibility, completeness and accuracy (e.g. see UNISDR-CRED (2018) related to EM-DAT database). These limitations hamper the validation of any large-scale modelling framework and I think they should be mentioned, either in section 4.3 or in a dedicated section. This is a very important issue. We agree that it is prudent to note this, although a long description is not possible due to space constraints. We propose to add the following to the start of section 4 (just before 4.1): "An overall challenge for global risk modellers is the lack of high-quality impact data for model validation. Efforts are constantly ongoing to improve the collection of impact data used in databases such as EM-DAT (CRED, 2020), NatCatService (Munich Re, 2020), DesInventar (UNDRR, 2020), and CATDAT (Daniell, 2020), but issues relating to incompleteness, fragmentation, bias, and differences in reporting conventions remain a challenge (e.g. Kron et al., 2012; CRED & UNISDR, 2018)."

(8) Table 1: please consider the idea of separating each hazard in a dedicated table. Thank you for the suggestion. We also considered this option in the original manuscript. However, we believe that providing the information for all hazards next to each other in one table provides a more simple reference point for comparing the different elements

across hazards. Therefore, we believe that changing this would weaken this valuable aspect of our current manuscript, and hence we would prefer to leave the table in its current format.

(9) Also, please define the meaning of IDF in the caption. Thank you for pointing out this omission. We have added this to the caption (IDF= intensity-damage function)

References cited by the reviewer or in our response to the reviewer • CRED, 2020. EM-DAT. The Emergency Events Database. Université catholique de Louvain (UCL) - CRED, Brussels, www.emdat.be • Daniell, J.E. 2020. CATDAT. The CATDAT Integrated Natural Catastrophes Database, Karlsruhe, http://www.risklayer.com/en/service/catdat/ • Kron, W., Steuer, M., Löw, P., Wirtz, A., 2012. How to deal properly with a natural catastrophe database – analysis of flood losses. Nat. Hazard. Earth Sys., 12, 535-550, doi:10.5194/nhess-12-535-2012 • Munich Re, 2020. NatCatSERVICE. Munich Re, Munich, https://natcatservice.munichre.com/ • Sampson, C.C., Smith, A.M., Bates, P.D., Neal, J.C., Alfieri, A., Freer, J.E., 2015. A high‐resolution global flood hazard model. Water Resour. Res., 51, 7358-7381, doi:10.1002/2015WR016954 • UNDRR, 2020. DesInventar database. UNDRR, Geneva, https://www.desinventar.net/ • UNISDR-CRED (2018). Economic Losses, Poverty and Disasters 1998-2017, https://www.unisdr.org/2016/iddr/IDDR2018_Economic%20Losses.pdf • Wing, O.E.J., Bates, P.D., Smith, A.M., Sampson, C.C., Johnson, K.A., Fargione, J., Morefield, P., 2018. Estimates of present and future flood risk in the conterminous United States. Environ. Res. Lett., 13, 034023, doi:10.1088/1748-9326/aaac65

---

## Author Comment (AC3) · 20 Feb 2020

Thank you for the opportunity to review this valuable & timely effort. For myself & my colleagues at the World Bank, this catalogue will be useful to support and employ risk analytics at global scale. For each of the hazards, the overview of the current state of disaster risk modeling is detailed and comprehensive, and the article's structure will make it a handy reference. We would like to thank the reviewer for the time taken to review our manuscript. We are delighted that it will be of value to organisations such as the World Bank, and very pleased that the referee states that the review is detailed and comprehensive. We respond to the specific comments of the review below.

(1) In the discussion of future research challenges & opportunities in section 4, I sug-

[Figure]

gest that the authors include as a subsection a brief mention of work on socio-economic vulnerability and resilience to disasters, at global level. Although socio-economic characteristics are typically left out of the canonical hazard/exposure/vulnerability heuristic (line 44), state-of-the-art risk analytics are increasingly moving to include them, as described by the authors throughout the text. For example, socio-economic heterogeneities are noted as relevant inputs to drought (lines 316, 344) and wildfire risk (lines 346, 356-8), and would likely also be useful to develop representations of landslides (lines 520-24). Deep interdependencies between these hazards' impacts and socio-economic risk factors may be one reason that modeling of these hazards is still relatively rudimentary, suggesting an opportunity for further research. Similarly, for hazards that have proved more tractable including flooding and earthquakes: the efficacy of DRR measures can depend greatly on socio-economic feedbacks including risk perception (lines 645, 753), social vulnerability (734), and resilience (740). Incorporating these datasets and dynamics will be essential to the development of CBA tools. As risk analytics achieve greater spatial resolution and theoretical sophistication, socioeconomic information layers are increasingly relevant for risk analytics, and essential for DRR applications (even apart from political economy considerations). For these reasons, and to summarize as an opportunity for research the point made throughout the text, I recommend that the authors include a brief note in Section 4. I know that the authors are familiar with this area from their own research, and I suggest as well my colleagues' work on the subject at global scale. Thank you. We agree with the reviewer on the importance of this issue. Some aspects were discussed in section 4.3, but based on the reviewer's comment we propose to further strengthen both section 4.3 (Vulnerability) to include some of the these important social vulnerability aspects as well as section 4.4 (DRR measures). The proposed text to be added to these 2 subsections reads as follows: Section 4.3 additional text: "Specifically, there is an increased recognition of the need to assess how socioeconomic processes can influence spatiotemporal changes in vulnerability (Cutter et al., 2015). For example, there can be a temporary rise in risk perception after a natural hazard, resulting in an increase in

DRR activities. Conversely, the absence of a natural hazard over a prolonged period can create a (false) sense of safety, which can increase vulnerability (Di Baldassarre et al., 2015). An improved understanding of these dynamics of socio-economic vulnerability can significantly improve the ability of risk managers to more efficiently implement DRR measures (Hallegatte et al., 2017; Wens et al., 2019). Recent studies have attempted to assess some of these aspects, for example by developing indicators of socioeconomic resilience for over 90 countries (Hallegatte et al., 2016), examining spatial differences in risk in different poverty groups (Hallegatte et al., 2015; Winsemius et al., 2018), or modelling dynamic feedbacks between levees and risk perception (Di Baldassarre et al., 2018; Haer et al., 2019). De Ruiter et al. (2020) discuss how the impacts of consecutive disasters can be distinctly different from single hazards due to changes in socioeconomic vulnerability" and "Moreover, international organisations on the ground are calling for an even higher level of granularity of these exposure, vulnerability and risk estimates in order to correctly target those individuals who are in mostly need of disaster relief aid. For this to be achieved, it is not only required to combine estimates of natural hazard with higher-resolution vulnerability and exposure information, but also to increase the level of detail of the latter for different groups, for example with regards to gender, income, livelihood, and access to healthcare." Section 4.4 additional text: "Another aspect that is often overlooked, especially on a global scale, is the interactions between different DRR measures that are aimed at specific hazards (Zaghi et al. 2016; Scolobig et al., 2017). DRR measures aimed at decreasing the risk of one hazard can increase the risk of another, so-called asynergies of DRR measures (De Ruiter et al., 2020). For example, building on stilts is a commonly used measure to decrease a building's flood vulnerability, but it can simultaneously increase a building's earthquake vulnerability (Wood and Good 2004). Accounting for such asynergies between DRR measures in a risk analysis is crucial, for example when developing tools that enable policy makers to assess the effectiveness of DRR measures. A first attempt to quantify these asymmetries at a large spatial scale has recently been carried out by De Ruiter et al. (2020), for measures to reduce flood and earthquake risk. The

expansion of these approaches to global scale would be a large step forward for global risk modelling."

(2) line 297 "is" misspelled as "us" Thank you. This will be amended

(3) lines 792 & 795 replace "between" with "among" Thank you. Both of these typos will be amended as suggested

References cited by the reviewer or in our response to the reviewer • De Ruiter, M.C., Couasnon, A., Van den Homberg, M.J.C., Daniell, J.E., Gill, J.C., Ward, P.J., 2020. Why we can no longer ignore consecutive disasters. Earths Future, online first, doi:10.1029/2019EF001425 • De Ruiter, M.C., De Bruijn, J.A., Englhardt, J., Daniell, J.W., Ward, P.J., De Moel, H., 2020. The asynergies of disaster risk reduction measures: comparing floods and earthquakes. In review. • Di Baldassarre, G., Viglione, A., Carr, G., Kuil, L., Yan, K., Brandimarte, L., Blöschl, G., 2015. Debates-Perspectives on socio-hydrology: Capturing feedbacks between physical and social processes. Water Resour. Res., 51, 4770-4781, doi:10.1002/2014WR016416 • Haer, T., Botzen, W.J.W., Aerts, J.C.J.H., 2019. Advancing disaster policies by integrating dynamic adaptive behaviour in risk assessments using an agent-based modelling approach. Environ. Res. Lett., 14, 044022, doi:10.1088/1748-9326/ab0770 • Hallegatte, S., Green, C., Nicholls, R.J., Corfee-Morlot, J., 2013. Future flood losses in major coastal cities. Nat. Clim. Change, 3, 802-806, doi:10.1038/nclimate1979 • Hallegatte, S., Bangalore, M., Bonzanigo, L., Fay, M., Kane, T., Narloch, U., Rozenberg, J., Treguer, D., Vogt-Schilb, A., 2016. Shock Waves: Managing the Impacts of Climate Change on Poverty. World Bank, Washington DC • Hallegatte, S., Vogt-Schilb, A., Bangalore, M., Rozenberg, J., 2017. Unbreakable: Building the Resilience of the Poor in the Face of Natural Disasters. Washington DC, World Bank • Scolobig, A., Komendantova, N., Mignan, A., 2017. Mainstreaming multi-risk approaches into policy. Geosciences, 7, 129, doi: 10.3390/geosciences7040129 • Ward, P.J., Jongman, B., Aerts, J.C.J.H., Bates, P.D., Botzen, W.J.W., Diaz Loaiza, A., Hallegatte, S., Kind, J.M., Kwadijk, J., Scussolini, P., Winsemius, H.C., 2017. A global framework for future costs and benefits of river-flood protection in urban areas. Nat. Clim. Chang., 7, 642-646, doi:10.1038/NCLIMATE3350 • Wens, M., Johnson, J.M., Zagaria, C., Veldkamp, T.I.E., 2019. Integrating human behavior dynamics into drought risk assessment—A sociohydrologic, agent‐based approach. WIREs Water, 6, e1345, doi:10.1002/wat2.1345 • Winsemius, H.C., Jongman, B., Veldkamp, T.I.E., Hallegatte, S., Bangalore, M., Ward, P.J., 2018. Disaster Risk, Climate Change, and Poverty: Assessing the Global Exposure of Poor People to Floods and Droughts. Environ. Dev. Econ., 23, 328-348, doi:10.1017/S1355770X17000444 • Wood, N.J, Good, J.W., 2004. Vulnerability of port and harbor communities to earthquake and tsunami hazards: the use of GIS in community hazard planning. Coast. Manage., 32, 243-269, doi:10.1080/08920750490448622 • Zaghi, A.E., Padgett, J.E., Bruneau, M., Barbato, M., 2016. Establishing common nomenclature, characterizing the problem, and identifying future opportunities in multihazard design. J. Struct. Eng., 142, H2516001, doi:10.1061/(ASCE)ST.1943-541X.0001586

———————————————